# HI-LIGHT: A PATH TO HIGH-FIDELITY, HIGH-RESOLUTION VIDEO RELIGHTING WITH A NOVEL EVALUATION PARADIGM

## ABSTRACT

Video relighting offers immense creative potential and commercial value but is hindered by challenges, including the absence of an adequate evaluation metric, severe light flickering, and the degradation of fine-grained details during editing. To overcome these challenges, we introduce Hi-Light, a novel, training-free framework for high-fidelity, high-resolution, robust video relighting. Our approach introduces three technical innovations: lightness prior anchored guided relighting diffusion that stabilises intermediate relit video, a Hybrid Motion-Adaptive Lighting Smoothing Filter that leverages optical flow to ensure temporal stability without introducing motion blur, and a LAB-based Detail Fusion module that preserves high-frequency detail information from the original video. Furthermore, to address the critical gap in evaluation, we propose the Light Stability Score, the first quantitative metric designed to specifically measure lighting consistency. Extensive experiments demonstrate that Hi-Light significantly outperforms state-of-the-art methods in both qualitative and quantitative comparisons, producing stable, highly detailed relit videos. Video demo can be found in the supplementary materials; the tested code is in the anonymous link below: https://anonymous.4open.science/r/Relight-Video-C666

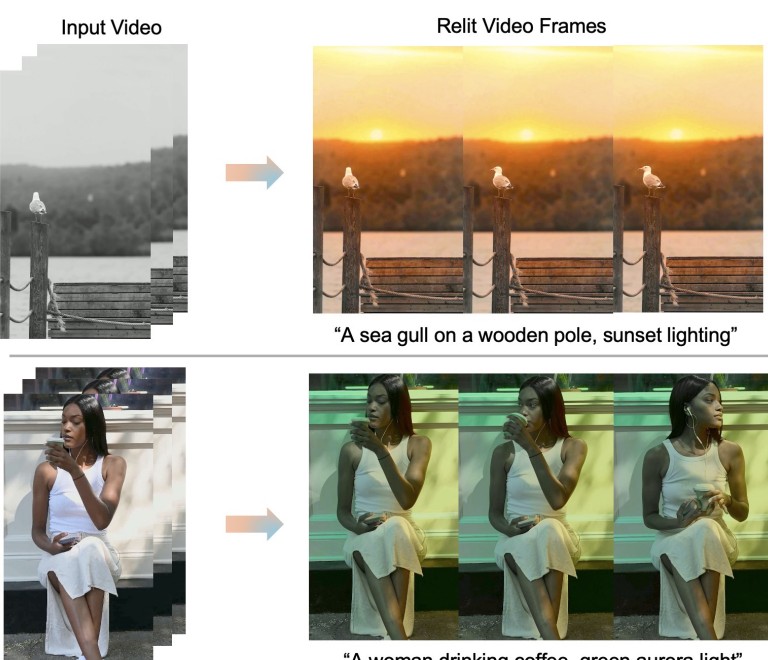

Figure 1: Demonstrations of the text-conditioned video relighting task by our framework.

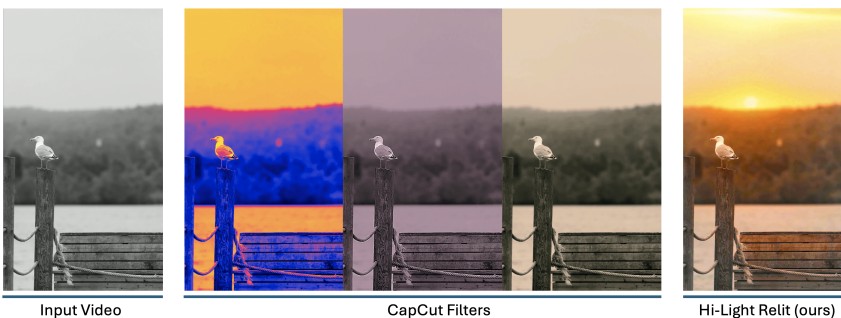

Figure 2: Comparison of the relighting effects of CapCut Sunset filters and our model.

# 1 INTRODUCTION

*"Lighting is the blank page. It's the canvas. It's the thing that you start with — you can't do anything until you have a light."* — Sir Roger Deakins

Lighting and its interaction with the environment shape how we perceive the world. Manipulating the light can dramatically alter the narrative and emotion buried in the visual media, making it a task of significant aesthetic and commercial value and a long-standing research problem in computer vision. However, video relighting remains a less explored and substantially more challenging domain. As shown in Figure 2, traditional video editing techniques, such as applying filters, are insufficient for this task. While these filters can perform global adjustments to colour and saturation, they cannot synthesize the complex, physically grounded effects of true illumination. Specifically, they fail to model crucial light properties such as directionality, highlights and shadows, or project lighting onto grayscale scenes effectively.

xDiffusion models have driven major progress in image generation and editing, spurring rapid advances in data-driven image relighting. Models such as IC-Light (Zhang et al., 2025) now define the state-of-the-art (SOTA). While image relighting leverages large-scale datasets with paired lighting conditions, such data is infeasible for video due to the cost of capturing dynamic scenes under multiple controlled illuminations. Light stage systems (Debevec et al., 2000) provide the closest alternative but require specialised equipment and capture primarily human subjects under simulated lighting, not real-world settings. This scarcity limits direct large-scale video training and necessitates more complex solutions. Existing approaches, including a recent breakthrough like Light-A-Video Zhou et al. (2025), still suffer from the following three weaknesses. **Detail degradation**: The reconstruction process within current video diffusion backbones is often imperfect, leading to a loss of high-frequency details. This causes sharp, fine-grained elements such as foliage and hair to become blurred or smoothed in the relit output. **Light flickering**: The existing solutions (Zhou et al., 2025; Liu et al., 2025) generally adopt IC-Light as the relighting model, and when single-image relighting models are applied on a per-frame basis, the resulting video often exhibits severe flicker. This lack of temporal coherence is a major artefact that jeopardises the quality of the final output. **Inadequate evaluation metrics**: Current evaluation methods are restricted to per-frame image metrics and subjective human evaluation. Standard metrics like FID and CLIP score are not able to capture the subtle degradation of fine-grained details. While human evaluation can be more accurate, its small sample size, potential for bias, and lack of scalability make it impractical for comprehensive benchmarking. Notably, there is no established metric designed to quantitatively measure the light flickering problem.

To tackle these challenges, we introduce Hi-Light, a training-free framework that decouples relighting from detail preservation. Hi-Light generates a smoothed, low-resolution relit video and then intelligently projects the new lighting onto the original high-resolution frames, preserving fine-grained details while ensuring light stability. Our framework's training-free design circumvents the need for large-scale datasets and costly training of new architectures by leveraging computer vision principles to surgically correct the common flickering and detail-loss artefacts of existing SOTA models. To address the evaluation gap, we also propose a new quantitative metric that jointly measures detail preservation and light stability. Our main contributions are:

- We present Hi-Light, a training-free, backbone-agnostic video relighting framework. Hi-Light is the only method capable of processing high-resolution video with computational efficiency, thereby extending accessibility to a wider community of users.

- We introduce a lightness-prior–anchored progressive fusion scheme that suppresses luminance oscillations during diffusion, and two plug-and-play modules: Hybrid Motion-Adaptive Light Smoothing Filter (HMA-LSF) to remove flicker, and LAB Detail-Preserving Fusion (LAB-DF) to restore fine-grained texture.

- We further propose the first principled evaluation paradigm for video relighting, including a Light Stability Score ($S_{LS}$) that complements standard fidelity metrics.

- As compared to the second-best method, our approach achieves an 80% improvement in light stability and a 56% improvement in detail preservation, setting the new SOTA.

## 2 RELATED WORK

**Visual Media Relighting**   Controlling illumination is a fundamental challenge in computer vision, with extensive research dedicated to single-image relighting. Early deep learning approaches made significant strides, particularly in portrait relighting (Nestmeyer et al., 2020; Kim et al., 2024). To tackle this, existing methods have explored various strategies. Some approaches rely on explicit 3D geometry and reflectance estimation. For example, SunStage (Wang et al., 2023) performs test-time optimization on a selfie video to reconstruct facial properties, while IllumiCraft (Lin et al., 2025) jointly models lighting and geometry using environment maps and 3D point tracks. Other works focus on disentanglement; LuminSculpt (Zhang et al., 2024) employs a network trained on synthetic data to decouple illumination from other scene factors. TC-Light (Liu et al., 2025) first aligns global exposure using a per-frame appearance embedding; subsequently, it refines fine-grained illumination and texture by optimizing a canonical representation called the Unique Video Tensor. Setting the new state-of-the-art, Light-A-Video (Zhou et al., 2025) recently introduced a training-free framework that adopts a progressive light fusion method to strengthen the temporal consistency in the denoising process.

**Reference-Based Video Quality Assessment**   For conditional generation tasks such as video-to-video translation or restoration, where a ground-truth reference exists, a variety of full-reference metrics are utilised. The most fundamental method is the Peak Signal-to-Noise Ratio (PSNR), which is based on simple mean squared error but often correlates poorly with human perception. A significant advancement was the Structural Similarity Index (SSIM), which provides a more perceptually relevant measure by comparing luminance, contrast, and structure (Wang et al., 2004). This was further improved by the Multi-Scale SSIM (MS-SSIM), which evaluates these properties across multiple resolutions for more robust results (Wang et al., 2003). To better align with human visual judgment, modern metrics leverage deep learning. The Learned Perceptual Image Patch Similarity (LPIPS) pioneered this by using features from deep networks to effectively predict perceptual similarity (Zhang et al., 2018). These assessments focused on the video details, while the lighting quality was generally missing; only small human evaluations have been done, but it is highly subjective and lacks consistency and scientific robustness (Zhou et al., 2025; Liu et al., 2025).

## 3 METHODOLOGY

### 3.1 RELIGHTING EVALUATION

A critical challenge in video relighting is the lack of an evaluation protocol that can simultaneously assess the two primary weaknesses of existing methods: temporal instability (light flickering) and the degradation of high-frequency details. To address this gap, we establish a new evaluation paradigm, Relit Video Evaluation. In the following subsections, we introduce a novel Light Stability Score to quantitatively measure light flickering and adopt the Structural Similarity Index (SSIM) to evaluate detail preservation.

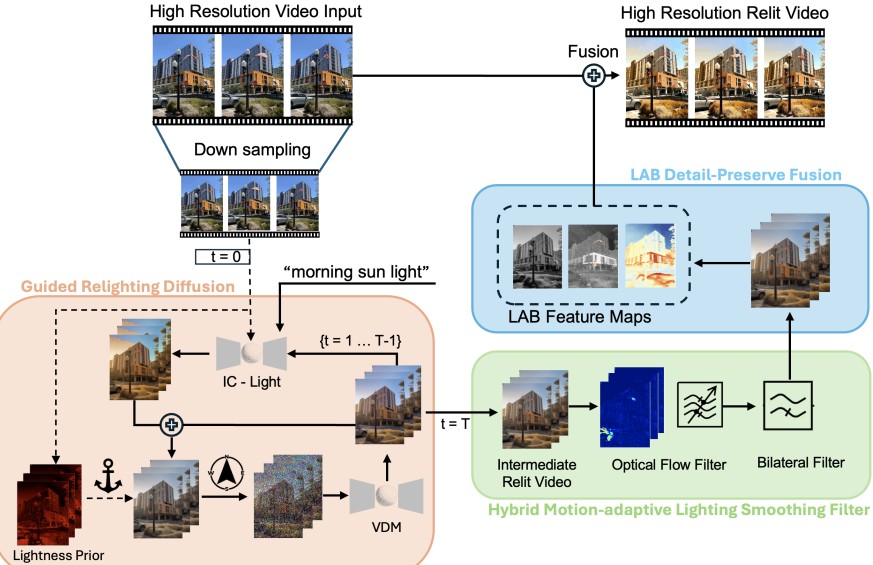

Figure 3: The overall structure of our Hi-Light framework. The framework first processes a down-sampled video through a guided relighting diffusion loop to generate lighting information where a lightness prior is anchored. The intermediate output is then stabilized using an HMA-LSF to eliminate flickering. Finally, the LAB-DF module transfers the illumination information original high-resolution source.

### 3.1.1 LIGHT STABILITY SCORE

The light flickering problem persists in all the existing work; therefore, it is important to propose a way to measure the stability of the lighting effect. First, to quantify temporal brightness dynamics from video data, each frame is converted to grayscale. A brightness threshold, $\tau$, is applied to segment each frame, yielding a set of suprathreshold pixels, $P_t$, which are the bright pixels that are susceptible to the flicker problem. Two time series signals are derived from this set: the average intensity of bright pixels $I_t$ and the number of bright pixels $C_t$. A third signal, representing the frame-to-frame change in average intensity, is derived by taking the first derivative of the $I_t$ series, $\dot{I}_t$.

Next, a quantitative smoothness score, $S$, is calculated for each of the three time series signals to assess the video's light fluctuation. This score is based on the magnitude of frame-to-frame changes relative to the signal's overall dynamic range. Given a series of video frames, $t = \{t_1, t_2, ..., t_N\}$, the mean absolute change, $M$, is first computed as $M = \frac{1}{N-1}\Sigma_{i=0}^{N-2}|t_{i+1} - t_i|$. This value is then normalized by the signal's peak-to-peak range, $R = max(t) - min(t)$, yielding a scale-invariant unsmoothness metric $U_{norm} = \frac{M}{R}$. Next, an exponential decay function transforms this metric into a final score, $S$, bounded by $0$ and $1$; a higher score denotes greater smoothness. Finally, the Light Stability Score, $S_{LS}$, will be the average of the scores: $S_{LS} = \frac{S_{I_t} + S_{C_t} + S_{\dot{I}_t}}{3}$.

### 3.1.2 DETAIL PRESERVATION

Existing relit videos suffer from detail loss during the diffusion process. To measure how many details are preserved in the relit video, we propose using the Structural Similarity Index (SSIM), which mimics how the human visual system works. Instead of just comparing individual pixels, SSIM evaluates the similarity of local patches of a frame based on three key components: luminance, contrast, and structure. The structure comparison is the most crucial component for evaluating the similarity of the details of videos; it compares the underlying shapes, textures, and patterns within

the patches. The formulation of SSIM is as follows:

$$Q(i,j) = \frac{2\mu_1(i,j)\mu_2(i,j) + C_1}{\mu_1^2(i,j) + \mu_2^2(i,j) + C_1} * \frac{2\sigma_1(i,j)\sigma_2(i,j) + C_2}{\sigma_1^2(i,j) + \sigma_2^2(i,j) + C_2} * \frac{\sigma_{12} + C_3}{\sigma_1(i,j) + \sigma_2(i,j) + C3}, \quad (1)$$

$$SSIM(I_1, I_2) = \frac{1}{MN}\Sigma_{i=1}^M \Sigma_{j=1}^N Q(i,j), \quad (2)$$

where $Q$ is the local quality score, $u_1$ and $u_2$ are the local means, $\sigma_1$ and $\sigma_2$ are the standard deviation, and $\sigma_{12}$ is the correlation of the frames $I_1$ and $I_2$, $C1$, $C_2$, and $C_3$ are the saturation constants. While multi-scale SSIM may perform better in general at the expense of computational resources and efficiency, we are only concerned with fidelity, regardless of colour, so SSIM is a better choice.

## 3.2 HI-LIGHT

To bridge this crucial gap between temporal coherence and visual fidelity, we introduce the Hi-Light video relighting framework as shown in the Figure 3. The video is first downsampled to $480p$ resolution. This step is necessary to utilize diffusion models trained at this resolution and reduces computational requirements (e.g., enabling inference on a single GPU). The downsampled video will then go through the guided relighting diffusion loop in the generation backbone to obtain an intermediate relit video, which possesses the relighting information but suffers from detail degradation and flickering light. The flicker light problem will then be handled by a Hybrid Motion-Adaptive Lighting Smoothing Filter, which is specifically designed to eliminate the flicker artifacts inherent in frame-by-frame generative processes. Finally, our novel LAB Detail-Preserve Fusion (LAB-DF) module intelligently transfers the stabilized lighting from the smoothed video onto the original high-resolution source video, preserving the high-frequency details.

### 3.2.1 PROGRESSIVE LIGHT FUSION GUIDED DIFFUSION WITH LIGHTNESS PRIOR ANCHORED

Directly applying an image relighting model results in inter-frame inconsistency. In the CIE LAB colour space (CIE, 1976), the $L$ channel encodes the perceptual lightness information — a monotone transform of scene luminance. Light flickering in the video relighting task manifests predominantly as low- to mid-frequency oscillations in the $L$ channel. Building on progressive light fusion (Zhou et al., 2025), we additionally inject a *per-step lightness prior* to damp luminance oscillations across the frames. We first define a high-pass lighting residual; let $I^{\text{in}}$ be the input video at $t=0$ which is a stable reference point. Define a static lightness residual by preserving the high-frequency information in the L channel of $I^{\text{in}}$ using a Gaussian filter $G_\sigma$:

$$\Delta L = L(I^{\text{in}}) - (G_\sigma * L(I^{\text{in}})), \qquad G_\sigma(x,y) = \frac{1}{2\pi\sigma^2}\exp\Big(-\frac{x^2 + y^2}{2\sigma^2}\Big). \quad (3)$$

The lightness residual $L - G_\sigma*L$ is preferred over raw gradients or a Laplacian transform because it is DC-insensitive and less noise-amplifying at very high frequencies due to the bounded gain $1 - \widehat{G_\sigma}(\omega)$, which is important for temporal stability. Then add the static lightness residual $\Delta L$ with empirically yielded fixed strength $\gamma = 0.3$:

$$L_t \leftarrow L_t + \gamma \Delta L, \quad (4)$$

which maintain a time-invariant anchor in the $L$ channel, reducing frame-to-frame variance in lighted regions and thus increasing stability. Because $\int \Delta^L \approx 0$ (mean-free under normalized $G_\sigma$), this addition minimally perturbs global brightness, avoiding drift.

At diffusion step $t$, we form the fused input as in progressive light fusion and denoise:

$$I_t^f = I_t^c + \lambda_t \Big(I_t^c - \tilde{I}_t^r\Big),$$
$$I_{t+1}^c = \mathcal{D}_\theta(I_t^f), \quad (5)$$

where the weight $\lambda_t$ is a gradually decreasing weight. This formulation initially emphasizes the influence of the relit target $\tilde{I}_t^r$, providing strong guidance in the early steps. As $\lambda_t$ decays, the process shifts toward refinement by the diffusion model, reducing dependency on the relit signal. The fused target at step $t$ is fed to the denoiser $\mathcal{D}_\theta(\cdot)$ to produce the next consistent state, thereby refining the denoising trajectory.

### 3.2.2 HYBRID MOTION-ADAPTIVE LIGHTING SMOOTHING FILTER (HMA-LSF)

A simple temporal smoother that averages frames may result in blurriness, especially for moving objects. To stabilise the light flickering problem, we design a novel hybrid temporal smoothing filter that adaptively integrates optical–flow–based motion compensation with bilateral filtering. Unlike prior temporal smoothers, which either blur motion or leave residual flicker, our design couples flow-based alignment with spatial edge-aware filtering, explicitly targeting the two dominant flicker sources: motion misalignment and compression noise.

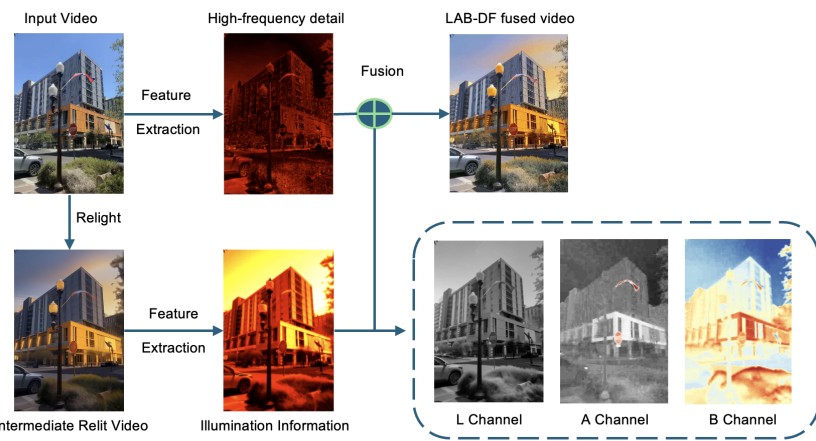

Figure 4: LAB feature maps will be extracted from the intermediate relit video and the input video. The high-frequency information from the input video will be fused with the illumination information from the intermediate relit video.

**Optical Flow Light Smoothing Filter** The main objective of the optical flow filter is to distinguish two types of brightness changes: Legitimate change, which is when an object gets brighter or darker, or a bright object moves to a new location, and illegitimate change (light flickering), where pixels' brightness fluctuates rapidly and erratically from one frame. Optical flow is the estimation of the motion of pixels between two consecutive frames. The filter will calculate a motion vector for every pixel in the frame, which is important for identifying moving objects. The equation for the optical flow constraint is given by:

$$I(x, y, t) = I(x + \delta x, y + \delta y, t + \delta t), \tag{6}$$

where $I(x, y, t)$ is the pixel intensity at position $(x, y)$ at time $t$ and the goal is to find the displacement $(\delta x, \delta y)$. Upon obtaining the flow, the filter performs motion compensation by taking the previously smoothed frame and warping it according to the calculated motion vectors. This warping aligns the content of the previous smoothed frame with the content of the current frame. The result is an estimate of what the previous smoothed frame would look like if the objects had moved to their new positions in the current frame. To address the flicker pixels, the filter blends the warped (motion-compensated) frame $f_{t-1}$ with the current frame $f_t$, and the blending is controlled by a weighted sum of the two frames as follows:

$$f_{blend} = \alpha f_{warped} + (1 - \alpha) f_t. \tag{7}$$

To avoid motion blur in fast-moving scenes, the $\alpha$ term is adaptive to the magnitude of motion. When motion is high, it reduces the $\alpha$ value, thereby relying more on the current frame and reducing the smoothing effect. This effectively prevents ghostly trails behind moving objects. Moreover, the change between two consecutive frames may be negligible, so we introduce a frame window; instead of just considering the immediately preceding frame, we calculate a weighted average of a window of frames, giving more importance to the most recent ones. This provides a more robust history of the pixel values, improving the smoother results.

Instead of adopting transformer-based optical flow models like RAFT Teed & Deng (2020), which demands over 80GB of VRAM for 81 video frames, we utilize the Farneback optical flow via the

OpenCV library. This CPU-only approach is computationally friendly, ensuring broader user accessibility.

**Bilateral Filter**   The bilateral filter is a non-linear, edge-preserving smoothing filter. For each pixel $p$, it calculates a weighted average of its surrounding pixels, and the weight depends on spatial distance and intensity difference:

$$BF[I]_p = \frac{1}{W_p} \Sigma_{q \in SW} G_{\sigma_s}(||p - q||) G_{\sigma_r}(|I_p - I_q|) \cdot I_q. \tag{8}$$

$p$ and $q$ are the target pixel and its neighbour pixels within the search window $SW$, $I$ denotes the intensity, $G$ is the Gaussian kernel, and $W_p$ is the normalisation factor that constrains the intensity value. A neighbouring pixel is assigned a low weight if it is either too far away or if its intensity is very different from the central pixel. When the filter is over a flat region, the intensity differences are small, so the range weights are high. The filter acts like a standard blur, smoothing out noise. When the filter crosses a sharp edge, the intensity differences become large. The weights for pixels on the other side of the edge drop to near zero, effectively ignoring them. This preserves the sharpness of the edge while still smoothing the areas on either side of it. To sum up, the optical flow filter tracks moving objects to smooth out inconsistent brightness fluctuations from one frame to the next and smooth the flicker intelligently. Then the bilateral filter reduces any remaining spatial noise, such as compression artefacts and inconsistent colour in flat regions.

### 3.2.3 LAB DETAIL-PRESERVE FUSION (LAB-DF)

As shown in Fig. 4, we convert frames to CIE LAB, where $L$ channel encodes perceptual lightness and $A$ and $B$ channels encode colour. A direct approach is copying the input's high-frequency $L$ into the relit video which does preserve details, but it introduces afterimages (ghosting). The afterimage problem is that the VDM can not reproduce geometry and edges exactly, so injecting fine-grained information from the input frame into a misaligned relit frame accumulates residuals.

To avoid afterimage, we invert the transfer: we take only the low-frequency illumination from the relit lightness and add it to the input lightness. Concretely,

$$L' = L_i + \beta \left( G_\sigma * L_r \right), \tag{9}$$

where $L_i$ and $L_r$ are the input and relit $L$ channels, $G_\sigma$ removes structural information from $L_r$, retaining exposure/contrast and light direction, and $\beta \in [0, 1]$ controls the transfer strength. We then combine the enhanced lightness with the relit chroma $A$ and $B$ channels to retain the intended colour of the new lighting:

$$V'(x, y) = \left[ L'(x, y), \, A_r(x, y), \, B_r(x, y) \right]. \tag{10}$$

This design effectively eliminates the afterimage problem, preserves the input's textures, and carries over the relit scene's colour and tonal style.

## 4 EXPERIMENT

**Baseline**   Given that the research in video generation and editing is relatively new, there is a shortage of video relighting work. We have conducted experiments using the open-source SOTA model Light-A-Video (Zhou et al., 2025) with its three different VDM backbones (CogVideoX (Yang et al., 2025), AnimateDiff (Guo et al., 2024), and Wan Wan et al. (2025)) and TC-Light (Liu et al., 2025). The configuration of the baseline models is the same as their demonstration configuration.

**Experiment Setup**   We conducted a rigorous comparative experiment where we used 100 video clips (70 human portraits, 30 non-human environments), collected from the internet and self-recorded. The video content spans diverse scenarios, including indoor and outdoor environments, as well as relatively static and highly dynamic scenes. The clips range from 1080p to 2160p and were standardised to 81 frames at 24 fps. Following the baselines, Hi-Light applies 30% noise to the input latent, with the VDM performing denoising over $T_m = 25$ steps to produce intermediate relit videos. The fusion weight is set as $\lambda_t = 1 - t/T_m$. Empirical tests determined amplification factors of 20, 20, and 5 for $S_{I_t}, S_{C_t}, S_{\hat{I}_t}$, respectively, to balance magnitudes. The hyperparameters were

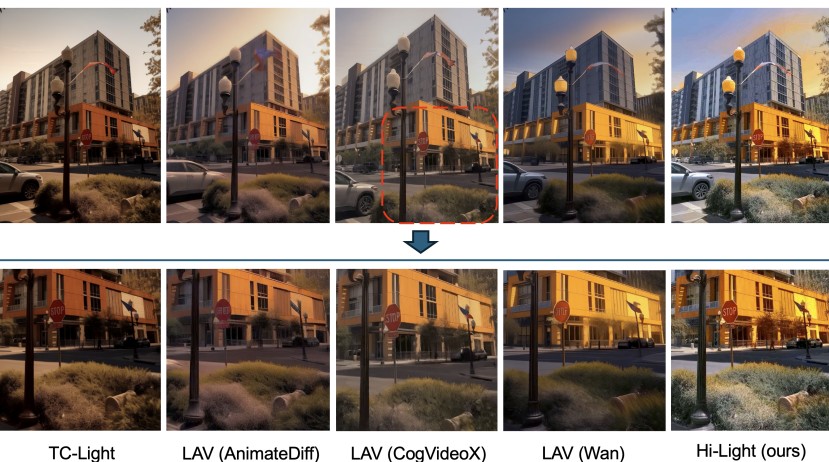

TC-Light          LAV (AnimateDiff)          LAV (CogVideoX)          LAV (Wan)          Hi-Light (ours)

Figure 5: A visual comparison of relighting methods with the text prompt "sunset lighting." Hi-Light achieves the best detail preservation. The top row shows a relit video frame, while the dotted red box marks regions of contrast, enlarged in the bottom row.

fixed for all the comparative experiments. The lighting prompts include: aurora light, reading lamp light, natural light, sunset light, morning light, dawn lighting, snowy winter lighting, light coming through a window, neon light, and torch flame light. The directions of light include: top, bottom, left, and right. The experiments were conducted using one L40 GPU.

## 5 RESULTS

We conduct a comprehensive evaluation of Hi-Light, comparing it against open-source SOTA methods across qualitative, quantitative, and signal-processing domains to validate its effectiveness. We first present a qualitative comparison in Figure 5, showcasing frames from a relit video. Competing methods like TC-Light produce a washed-out effect, while the LAV variants suffer from significant detail degradation, resulting in blurry textures on the building and foreground foliage. In contrast, Hi-Light renders the new lighting with sharp highlights, while preserving the original high-frequency details. To quantify these visual improvements, we evaluate all methods using SSIM for detail preservation and our proposed Light Stability Score for temporal coherence, with results shown in Figure 6 and Table 1. The scatter plot provides an intuitive visualization of Hi-Light's superior performance. Our method is positioned in the top-right quadrant, indicating both high detail fidelity and robust lighting stability. It is notably close to the original input video, which represents the ideal target for these metrics. In contrast, all competing methods are clustered in a region corresponding to significantly lower SSIM and stability scores, highlighting their trade-off between applying new lighting and preserving the high-frequency detail. The accompanying table offers a detailed numerical breakdown that reinforces this conclusion. Hi-Light achieves an SSIM of 0.943, significantly outperforming the next best method, LAV (Wan), which scores 0.604. This large margin numerically highlights the exceptional detail preservation capability of our framework. Furthermore, Hi-Light attains an overall Light Stability Score of 0.470, substantially surpassing all competitors, achieving the highest scores across all components. This quantitative evidence conclusively demonstrates that Hi-Light sets a new SOTA standard, uniquely capable of achieving stable, high-fidelity video relighting without compromise.

To further understand the underlying reasons for this performance, we analyze the outputs in the frequency and temporal domains (Figure 7). The frequency analysis on the left provides strong evidence for Hi-Light's detail preservation capability. The Fourier spectra of competing methods reveal a pronounced attenuation of high-frequency components, visually confirming a loss of sharpness. In contrast, the spectrum from Hi-Light is nearly indistinguishable from the input's, proving that our method relights the scene without sacrificing textural information. Simultaneously, the light stability plots on the right validate our approach to eliminating flicker. The "Frame-to-Frame Change in Avg. Bright Pixel Intensity" graph is particularly revealing; competitors exhibit highly erratic

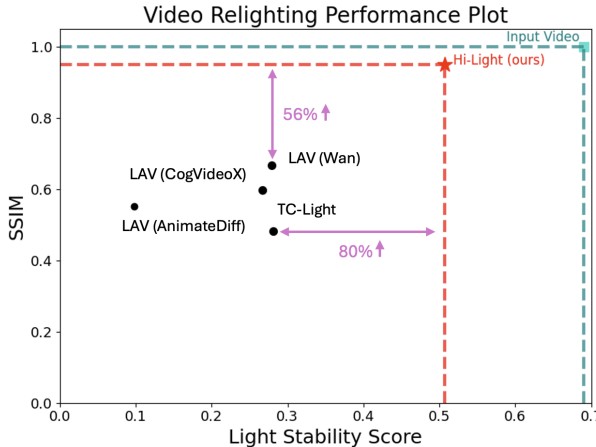

Figure 6: Visualized comparison of methods. Hi-Light outperforms other methods in both SSIM and Light Stability Score, achieving relative improvements of 56% and 80%, respectively, compared to the second-best methods.

| Model | SSIM ($\uparrow$) | LPIPS ($\downarrow$) | $S_I$ ($\uparrow$) | $S_c$ ($\uparrow$) | $S_j$ ($\uparrow$) | $S_{LS}$ ($\uparrow$) |
|---|---|---|---|---|---|---|
| TC-Light (Liu et al., 2025) | 0.484 | 0.464 | 0.283 | 0.184 | 0.376 | 0.281 |
| LAV (AnimateDiff) (Zhou et al., 2025) | 0.552 | 0.434 | 0.052 | 0.041 | 0.202 | 0.098 |
| LAV (CogVideoX) (Zhou et al., 2025) | 0.597 | 0.402 | 0.147 | 0.242 | 0.380 | 0.267 |
| LAV (Wan) (Zhou et al., 2025) | 0.604 | 0.395 | 0.119 | 0.281 | 0.437 | 0.279 |
| Hi-Light (ours) | **0.943** | **0.247** | **0.572** | **0.537** | **0.417** | **0.509** |

Table 1: Quantified results of the video relighting models.

fluctuations, a quantitative signature of severe flicker artefacts. Conversely, Hi-Light maintains a remarkably more stable and smooth profile. These plots directly demonstrate the efficacy of our hybrid motion-adaptive smoothing filter in producing temporally coherent results.

To validate the perceptual relevance of our proposed $S_{LS}$, we conducted a comprehensive human evaluation with 30 participants, including a computer vision professor, seven professionals from the digital art industry, and 22 graduate students. In a blind-test protocol designed to avoid bias, each participant was asked to rank three sets of videos based on light stability and video detail quality. Each set contained videos generated by the comparative models. Hi-Light was ranked as the top-performing model in 95.6% of the evaluations for light stability and 91.1% for video quality. The results demonstrate the superiority of our method and the soundness of $S_{LS}$.

# 6 ABLATION STUDY

| Method | | | SSIM ($\uparrow$) | $S_I$ ($\uparrow$) | $S_c$ ($\uparrow$) | $S_j$ ($\uparrow$) | $S_{LS}$ ($\uparrow$) |
|---|---|---|---|---|---|---|---|
| LAB-DF | HMA-LSF | Lighting Prior | | | | | |
| ✗ | ✗ | ✗ | 0.607 | 0.165 | 0.259 | 0.430 | 0.285 |
| ✗ | ✗ | ✓ | 0.615 | 0.205 | 0.321 | 0.533 | 0.353 |
| ✗ | ✓ | ✗ | 0.612 | 0.425 | 0.450 | 0.511 | 0.462 |
| ✗ | ✓ | ✓ | 0.623 | 0.385 | 0.495 | 0.534 | 0.476 |
| ✓ | ✓ | ✓ | **0.943** | **0.572** | **0.537** | **0.417** | **0.509** |

Table 2: Ablation study result on our methodology.

Starting from the baseline, adding the Lighting Prior alone raises stability modestly with a small SSIM change. Its main effect is on the derivative term $S_j$, showing that it damps frame-to-frame light changes. Adding the Smooth Filter on top of the prior delivers the largest stability gain, especially

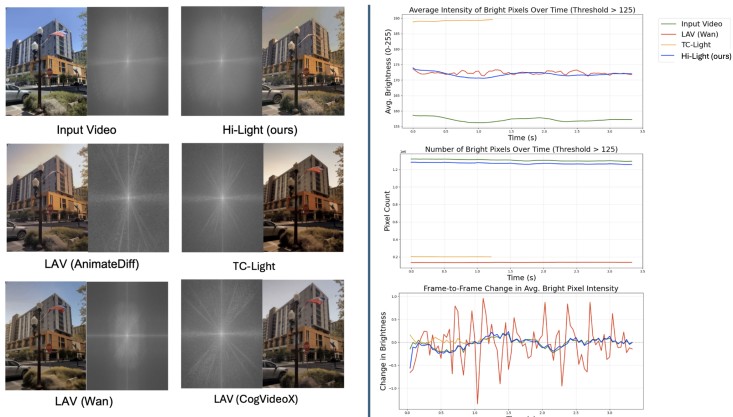

Figure 7: **Left** shows the frequency magnitude spectra. Our edited video has an identical-looking spectrum to the original video, suggesting that it retains most of the fine-grained details. Meanwhile, the other baselines have a more concentrated spread resulting from detail degradation. **Right** shows the three smoothness scores. Plot for TC-Light is shorter as it has 30 frames. For plot clarity, only LAV (Wan) is included here; the complete plots can be found in Appendix A.1.

in $S_I$ and $S_c$, pushing $S_{LS}$ to 0.462, while fidelity stays similar—this module is the key driver of temporal smoothness. Pairing the Lighting Prior with LAB-DF instead yields a large jump in SSIM (0.939) with moderate stability gains, confirming that LAB-DF is the chief detail-preserving component. Combining all three gives the best of both: the highest SSIM and the strongest overall stability. Appendix A.1 contains a comprehensive and detailed ablation study touching on efficiency, the VDM backbone, the number of time steps and noise strength.

## 7    CONCLUSION

In this work, we addressed the key challenges of detail degradation and the light flickering problem in video relighting. We introduced Hi-Light, a novel training-free framework that successfully generates high-fidelity and flicker-free relit videos. Our primary contributions include a lightness prior anchored diffusion scheme and a novel HMA-LSF that ensures temporal coherence and a LAB-DF module that preserves fine-grained details with remarkable fidelity. We also proposed the Light Stability Score, a new quantitative metric to standardise the evaluation of lighting stability, a critical but previously overlooked aspect of the task. Through comprehensive experiments, we have shown that Hi-Light significantly surpasses existing SOTA methods, establishing a new benchmark for quality and robustness in video relighting and opening up new possibilities for creative video editing. Our methodology can be extended to broader video editing by anchoring task-specific attribute priors (e.g. texture, style, colour), applying the same progressive residual to edit a long, temporally consistent video.

## 8    LIMITATION AND FUTURE WORK

While our framework achieves SOTA performance in video relighting, we observe a limitation common to current approaches: the suppression of lighting in high-contrast scenes. Specifically, if the input video contains saturated highlights or strong directional lighting, the model struggles to fully disentangle these features, resulting in a limited relighting effect. Furthermore, achieving fine-grained control over specific illumination components, such as the precise manipulation of shadows and highlights, remains an open challenge. Future work will focus on improving the disentanglement of intrinsic lighting and integrating more explicit controls for shadow synthesis.

## 9 Reproducibility Statement

We have made significant efforts to ensure reproducibility. The main paper details the architecture of Hi-Light, the Light Stability Score formulation, and all baseline configurations. Appendix A provides a detailed experimental setup and comprehensive ablation studies on architecture design, backbones, noise levels, and time steps, along with runtime analyses. Code, scripts, and instructions for reproducing all results are tested and provided via the anonymous repository link (https://anonymous.4open.science/r/Relight-Video-C666). Together, these resources should enable independent verification of all reported findings.

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

## A  APPENDIX

### A.1  COMPREHENSIVE ABLATION STUDY

### A.1.1  ABLATION STUDY ON THE ARCHITECTURE DESIGN

| Method | | | SSIM ($\uparrow$) | $S_I(\uparrow)$ | $S_c(\uparrow)$ | $S_{\tilde{I}}(\uparrow)$ | $S_{LS}(\uparrow)$ |
|---|---|---|---|---|---|---|---|
| LAB-DF | HMA-LSF | Lightness Prior | | | | | |
| ✗ | ✗ | ✗ | 0.607 | 0.165 | 0.259 | 0.430 | 0.285 |
| ✗ | ✗ | ✓ | 0.615 | 0.205 | 0.321 | 0.533 | 0.353 |
| ✗ | ✓ | ✗ | 0.612 | 0.425 | 0.450 | 0.511 | 0.462 |
| ✗ | ✓ | ✓ | 0.623 | 0.385 | 0.495 | 0.534 | 0.476 |
| ✓ | ✗ | ✗ | 0.939 | 0.295 | 0.400 | 0.475 | 0.390 |
| ✓ | ✗ | ✓ | 0.941 | 0.370 | 0.522 | 0.545 | 0.479 |
| ✓ | ✓ | ✗ | 0.942 | 0.549 | 0.523 | 0.373 | 0.482 |
| ✓ | ✓ | ✓ | **0.943** | **0.572** | **0.537** | **0.417** | **0.509** |

Table 3: A Complete Ablation Study Result on the Architecture of Hi-Light.

It appears that the combination of LAB-DF and Lightness Prior has a negative impact on $S_{\tilde{I}}$. A possible explanation is that LAB-DF restores edges in the L channel, so the local gradients get steeper inside bright areas. The HMA-LSF reduces boundary flips, but any residual flow error causes small intensity shifts within the bright set. Together, they may amplify tiny alignment errors inside bright regions, affecting the $S_{\tilde{I}}$.

| Module | Time taken (s) |
|---|---|
| Lightness Prior | 98 |
| HMA-LSF | 117 |
| LAB-DF | 7 |

Table 4: The table contains the average time taken for executing each module in the architecture.

As shown in Table 4, the Lightness Prior and HMA-LSF modules require around 117 seconds each, while the LAB-DF module executes much faster at only 7 seconds on average. The execution time of HMA-LSF scales strongly with resolution because both optical flow estimation and bilateral filtering are dense, pixel-level operations. Optical flow must compute and apply motion vectors for every pixel between frames, while bilateral filtering evaluates spatial–intensity neighbourhoods for each pixel to preserve edges. As resolution increases, the number of operations grows quadratically, leading to significantly higher runtimes for high-resolution videos compared to lower-resolution inputs. Also, increasing the denoising time step will increase the time taken for the lightness prior anchored and guided diffusion process.

Figure 9 shows an example of the effects of the introduced light smoothing filter. The filter mitigates the light flickering problem by stabilizing the video's lighting over time. The plots for "Average Intensity" and "Number of Bright Pixels" show that the unsmoothed video (red line) exhibits large and erratic fluctuations. In contrast, the smoothed video (green line) displays much more consistent and stable values for these metrics. The most direct evidence is in the "Frame-to-Frame Change" graph. The unsmoothed video shows frequent, high-amplitude spikes, which quantitatively represent severe flicker. The smoothed video, however, maintains a line consistently closer to zero, indicating that the change in brightness between consecutive frames is smooth.

Besides the framework architecture ablation study, we also investigated how the fusion strength affects the LAB-DF process. As shown in Figure 10, as the fusion strength increases, more weight is assigned to the $L$ channel of the smoothed intermediate relit video, resulting in a decrease in both SSIM and the light stability score. Notably, the decrease in SSIM appears to be almost linear, but there is a relatively sharp drop in the Light Stability Score when the fusion strength increases from 0.3 to 0.5.

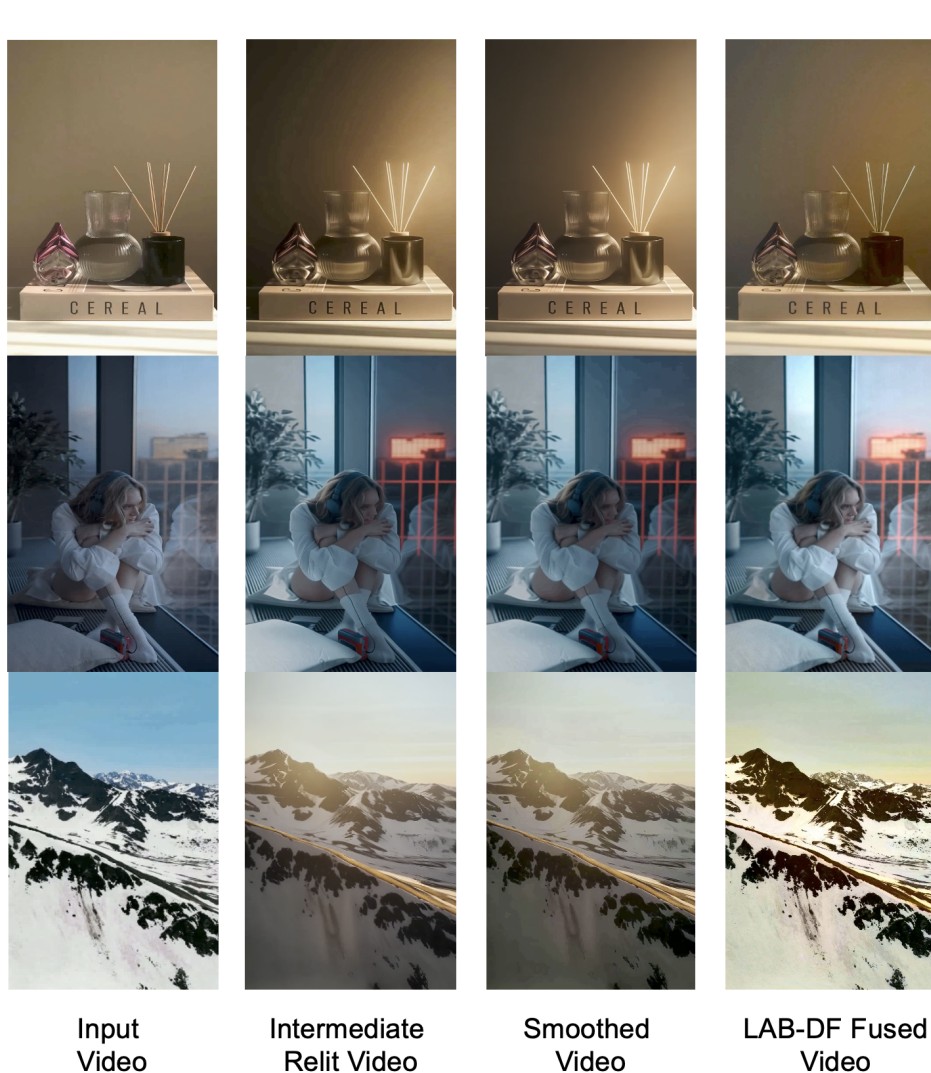

| Input
Video | Intermediate
Relit Video | Smoothed
Video | LAB-DF Fused
Video |

Figure 8: A visualization of the ablation study of the architecture.

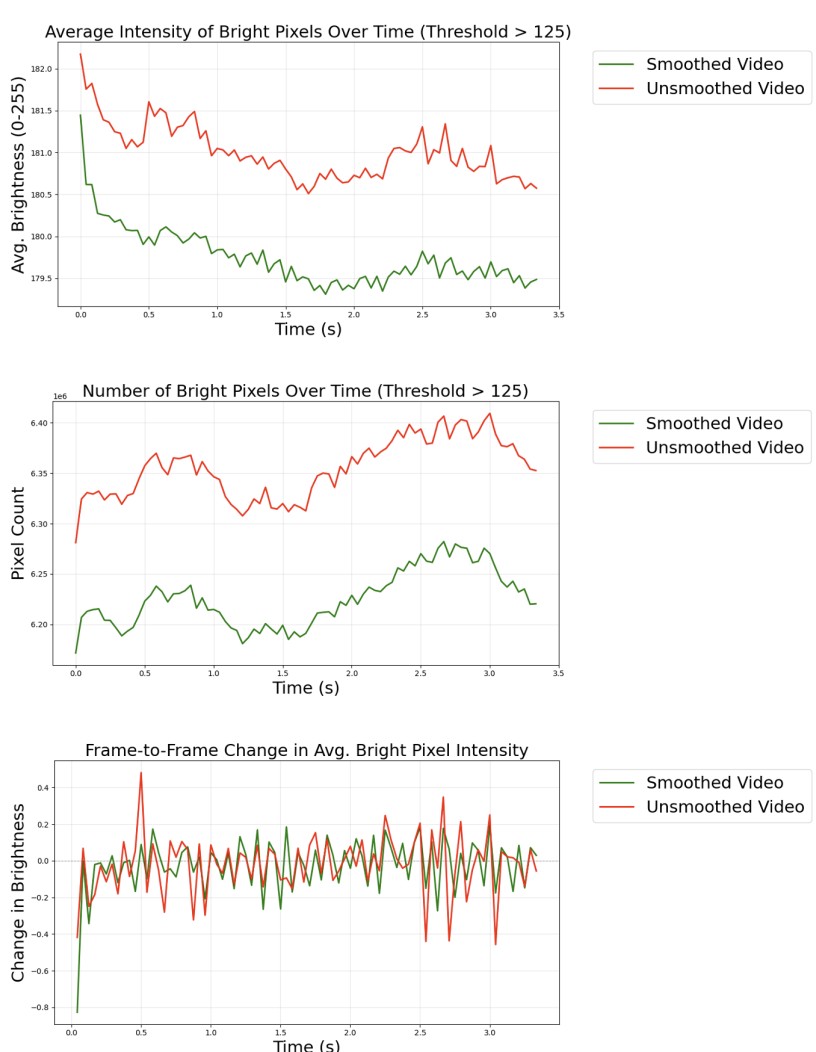

Figure 9: The plot shows the effect of the hybrid motion-adaptive light smoothing filter.

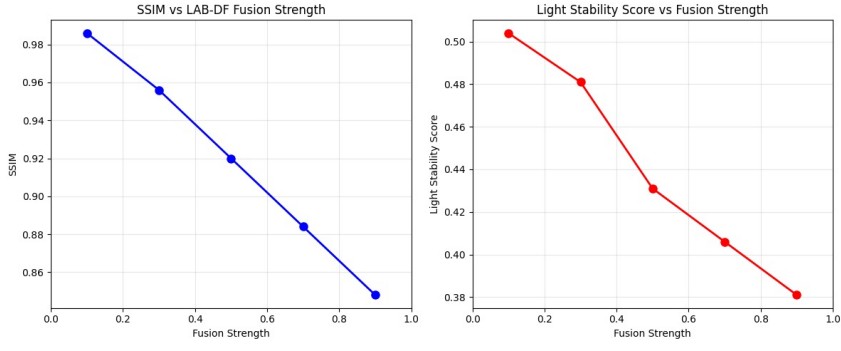

Figure 10: The plot shows how the strength of LAB-DF fusion affects SSIM and Light Stability Score.

A.1.2 ABLATION STUDY ON THE VIDEO DIFFUSION MODEL BACKBONE

| VDM | SSIM (↑) | $S_I$ (↑) | $S_c$ (↑) | $S_{\hat{j}}$ (↑) | $S_{LS}$ (↑) | Ave. Time Taken (s) |
|---|---|---|---|---|---|---|
| AnimateDiff | 0.749 | 0.360 | 0.228 | 0.389 | 0.326 | 371 |
| CogVideoX | 0.932 | 0.581 | 0.459 | 0.509 | 0.517 | 966 |
| Wan | 0.940 | 0.606 | 0.461 | 0.517 | 0.528 | 712 |

Table 5: Quantitative results for the ablation study on the VDM backbones.

To determine the optimal VDM backbone for our Hi-Light framework, we conducted a rigorous ablation study evaluating three open-source models: AnimateDiff (Guo et al., 2024), CogVideoX (Yang et al., 2025), and Wan (Wan et al., 2025). There are a total of 30 videos involved with a fixed time step of 10, and noise strength of 0.3. The quantitative results, presented in Table 5, reveal a distinct trade-off between computational efficiency and the quality of the intermediate video generated. The Wan backbone achieved superior performance, attaining the highest scores in both detail preservation and light stability. CogVideoX followed closely in quality but at a significant computational cost, requiring over 35% more processing time. Conversely, while AnimateDiff offered the fastest inference, it did so at the expense of a substantial degradation in both structural fidelity and light stability. Based on this analysis, Wan emerges as the most balanced choice. Therefore, we adopt Wan as the default VDM backbone for comparative experiments.

A.1.3 ABLATION STUDY ON THE TIME STEP

| # of Time Step | SSIM (↑) | $S_I$ (↑) | $S_c$ (↑) | $S_{\hat{j}}$ (↑) | $S_{LS}$ (↑) | Ave. Time Taken (s) |
|---|---|---|---|---|---|---|
| 5 | 0.952 | 0.590 | 0.407 | 0.506 | 0.501 | 617 |
| 10 | 0.956 | 0.600 | 0.467 | 0.609 | 0.559 | 771 |
| 15 | 0.951 | 0.599 | 0.472 | 0.591 | 0.554 | 1044 |
| 20 | 0.956 | 0.607 | 0.510 | 0.594 | 0.570 | 1258 |
| 25 | 0.952 | 0.607 | 0.492 | 0.598 | 0.566 | 1477 |

Table 6: Quantified results for the ablation study on the number of time steps.

To study the effect of the number of denoising time steps in our framework, which governs the fundamental trade-off between light stability and computational cost. We performed a rigorous ablation study on a sample of 30 videos, with results presented in Table 7. Our analysis reveals a non-monotonic relationship between the number of steps and the final output quality. While computational cost scales linearly, the Light Stability Score $S_{LS}$ improves from 0.501 to a peak of 0.570 at 20 steps, a 13.9% gain in the light stability. Beyond this point, however, we observe diminishing returns; increasing to 25 steps not only incurs a substantial additional runtime of over 200 seconds but also results in a slight degradation of both stability ($S_{LS}$ drops to 0.566) and detail preservation. These empirical results lead us to a principled choice, identifying 20 time steps as the optimal setting for our framework, as it maximises temporal coherence without introducing unnecessary computational overhead or performance degradation. Notably, 10 steps can also yield a quality result in a much shorter time.

A.1.4 ABLATION STUDY ON THE NOISE ADDITION

The ablation study on noise strength, conducted over 30 videos, shows a clear trade-off between structural fidelity and lighting stability. As noise strength increases from 0.1 to 0.6, SSIM decreases gradually, indicating progressive loss of fine-grained details. Meanwhile, the Light Stability Score also drops, reflecting reduced temporal smoothness. Interestingly, moderate noise levels (0.3–0.4) yield a balance, where the light stability scores remain relatively high while detail preservation is only slightly degraded. However, beyond 0.5, both stability and fidelity decline sharply, suggesting over-noising destabilises the diffusion process.

Although lower noise strengths tend to produce more stable and detail-preserving results, they can also limit the diversity of relighting outcomes by keeping the latent space of the relit video too close to the original input. Increasing the noise strength injects additional stochasticity into the diffusion process, enabling the model to explore a broader range of illumination effects and generate more

| Noise Strength | SSIM ($\uparrow$) | $S_I$ ($\uparrow$) | $S_c$ ($\uparrow$) | $S_{\dot{\imath}}$ ($\uparrow$) | $S_{LS}$ ($\uparrow$) |
|---|---|---|---|---|---|
| 0.1 | 0.937 | 0.674 | 0.497 | 0.486 | 0.552 |
| 0.2 | 0.936 | 0.635 | 0.492 | 0.492 | 0.540 |
| 0.3 | 0.934 | 0.609 | 0.489 | 0.492 | 0.530 |
| 0.4 | 0.932 | 0.642 | 0.471 | 0.490 | 0.534 |
| 0.5 | 0.930 | 0.609 | 0.475 | 0.527 | 0.537 |
| 0.5 | 0.925 | 0.572 | 0.500 | 0.462 | 0.511 |
| 0.6 | 0.923 | 0.500 | 0.462 | 0.465 | 0.476 |

Table 7: Quantified results for the ablation study on the diffusion noise strength.

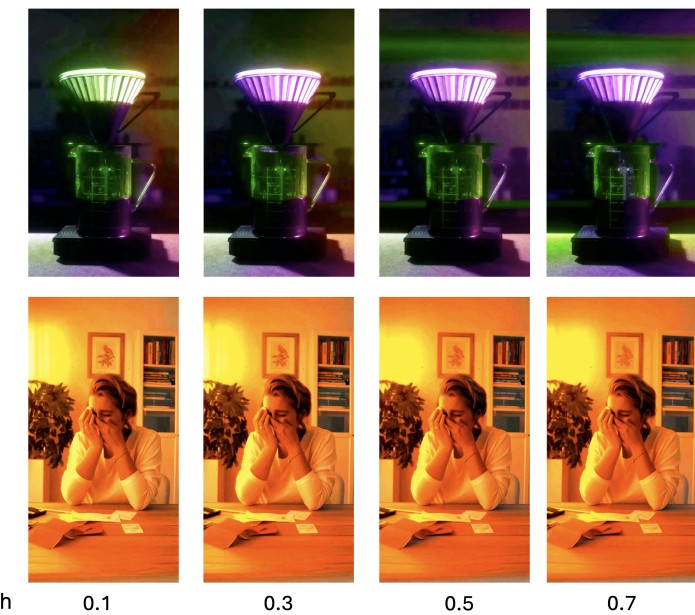

Noise Strength     0.1     0.3     0.5     0.7

Figure 11: A visualisation of the effect of noise strength.

varied relighting results as shown in Figure 11. However, this comes at the cost of reduced structural fidelity and light stability, highlighting an inherent trade-off. In practice, noise strength thus becomes a tunable option that users can adjust depending on whether they prioritise stability and fidelity or prefer more diverse lighting variations.

## A.2   EMPIRICAL CHOICE OF THE BRIGHTNESS THRESHOLD $\tau$ IN $S_{LS}$ CALCULATION

The brightness threshold $\tau = 125$ in the calculation of $S_{LS}$ is an empirical selection. As shown in the Figure 12, the distribution of the relit video clearly shifts toward higher brightness compared to the original, with a pronounced increase in pixel density beginning around L = 120 to 130. Below this range, both original and relit distributions overlap considerably, indicating that fluctuations are more likely due to scene content or motion rather than relighting. By contrast, above 125, the relit histogram diverges strongly from the original, capturing the excess brightness introduced by relighting. Choosing 125, therefore, strikes a balance: it is high enough to exclude darker regions where variations are unrelated to illumination changes, while low enough to consistently include the portion of the distribution where relighting effects dominate. This cutoff ensures that the $S_{LS}$ focuses on the visually meaningful relighted regions without being contaminated by background fluctuations and motion.

The Figure 13 also illustrates the effect of different luminance thresholds (100, 125, 150) on isolating bright regions within a frame. At a low threshold of 100, the mask includes a large portion of the

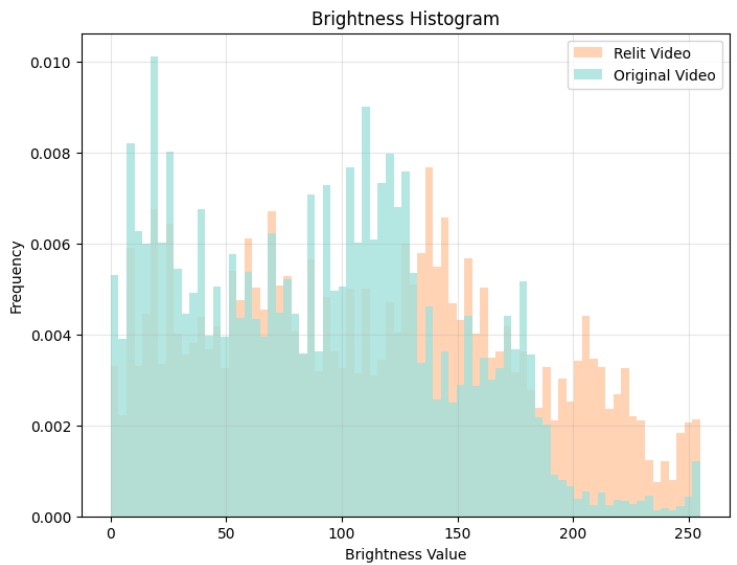

Figure 12: Histogram of pixel brightness of source videos and relit videos.

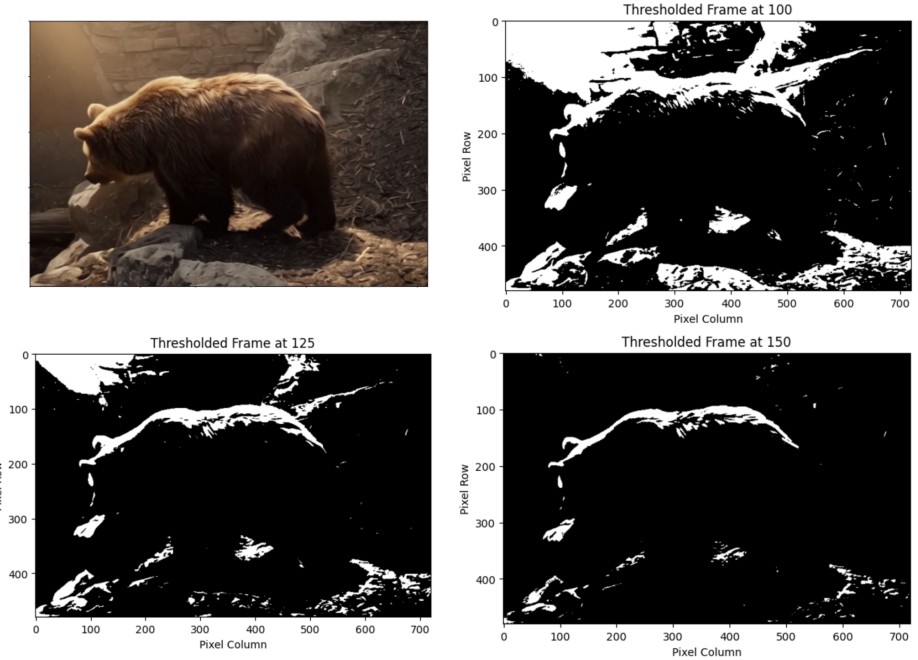

Figure 13: Visualisation of a relit video frame at different brightness.

| Model | 105 | 115 | 125 | 135 | 145 |
|---|---|---|---|---|---|
| TC-Light | 0.300 | 0.296 | 0.285 | 0.277 | 0.238 |
| LAV (AnimateDiff) | 0.119 | 0.123 | 0.109 | 0.115 | 0.120 |
| LAV (CogVideoX) | 0.202 | 0.208 | 0.209 | 0.231 | 0.239 |
| LAV (Wan) | 0.263 | 0.278 | 0.271 | 0.272 | 0.270 |
| Hi-Light | **0.515** | **0.509** | **0.520** | **0.521** | **0.536** |

Table 8: Sensitivity test results for the brightness threshold $\tau$. The table shows the $S_{LS}$ under different $\tau$ values.

background and textured regions, potentially introducing noise unrelated to relighting. At the high threshold of 150, only a few sparse highlights remain, missing much of the relevant relit areas on the bear's back and surrounding surfaces. In contrast, the threshold of 125 yields a relatively balanced mask: it successfully captures the prominent illuminated regions (e.g., along the bear's contour and the lit rock surfaces) while excluding most of the darker, non-relit areas. This visually demonstrates that 125 is high enough to avoid contamination from shadow and texture variation, yet good enough to preserve the meaningful bright zones where relighting actually occurs.

### A.2.1 SENSITIVITY STUDY ON THE BRIGHTNESS THRESHOLD $\tau$

To examine the robustness of the proposed $S_{LS}$ against variations in the brightness threshold $\tau$, we conducted a sensitivity analysis across values ranging from 105 to 145. As shown in Table 8, the relative ranking of models remains consistent regardless of the chosen threshold, with Hi-Light achieving the highest scores across all cases. While small numerical fluctuations are observed as $\tau$ varies, the overall performance gap between Hi-Light and competing methods remains substantial. This consistency indicates that $S_{LS}$ is not overly sensitive to the precise threshold choice and that our evaluation reliably captures temporal lighting stability. Importantly, the stability of Hi-Light improves slightly at higher thresholds, suggesting that the framework not only generalises across different luminance regimes but also benefits when the evaluation focuses on regions with more prominent illumination. Together, these findings confirm that our metric and results are robust and not an artefact of the specific choice of $\tau$.

### A.3 HUMAN SURVEY RESULT ANALYSIS

To quantitatively substantiate the perceptual relevance of our proposed Light Stability Score, we performed a Spearman's rank correlation analysis between our metric's rankings and the results of a 30-participant human evaluation. As shown in Figure 14 and Table 9, the analysis yielded a perfect positive correlation (Spearman's $\rho = 1.0$), indicating that the model rankings produced by our metric are identical to those derived from human judgment. This statistically significant alignment provides strong empirical evidence that the Light Stability Score serves as an effective and reliable proxy for human perception of light stability in the video relighting task.

| Model | $S_{LS}$ | $S_{LS}$ Rank | Ave. Human Rank | Human Rank |
|---|---|---|---|---|
| Hi-Light (ours) | 0.509 | 1 | 1.14 | 1 |
| TC-Light (Liu et al., 2025) | 0.281 | 2 | 2.59 | 2 |
| LAV (Wan) (Zhou et al., 2025) | 0.279 | 3 | 2.92 | 3 |
| LAV (CogVideoX) (Zhou et al., 2025) | 0.267 | 4 | 3.86 | 4 |
| LAV (AnimateDiff) (Zhou et al., 2025) | 0.098 | 5 | 4.50 | 5 |

Table 9: Comparison between the results of Light Stability Score and human survey.

### A.4 THE USE OF LARGE LANGUAGE MODEL

Large language models were used for grammatical error correction, LaTeX format correction, and debugging.

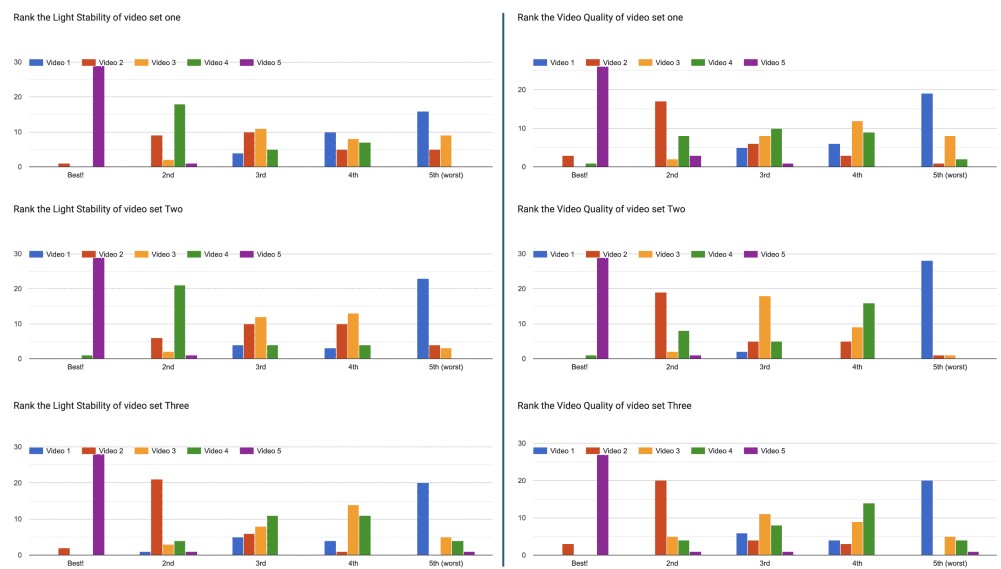

Figure 14: The left side shows the results of the human survey for light stability. The right side shows the results of the human survey for video quality. Video 1: LAV-AnimateDiff, Video 2: LAV-Wan, Video 3: LAV-CogVideoX, Video 4:TC-Light, Video 5:Hi-Light.

## A.5    REPRODUCIBILITY

The proposed framework Hi-Light can be easily reproduced by following the codes and instructions in the anonymous link in the abstract.

## A.6    ADDITIONAL PLOT RESULTS AND SHOWCASES

This section provides more visual results for the paper. The plots below provide a clear visual comparison of the light stability of the relit results. Hi-Light (ours) demonstrates the best smoothness in the plots among all the baselines.

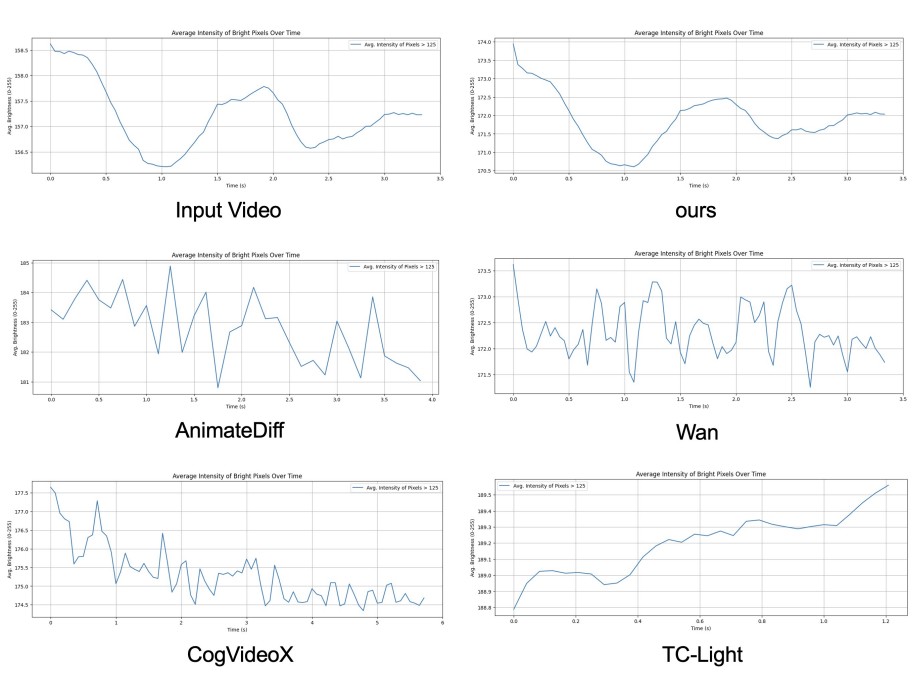

Figure 15: Plot of average intensity of bright pixels (brightness≥125) over time.

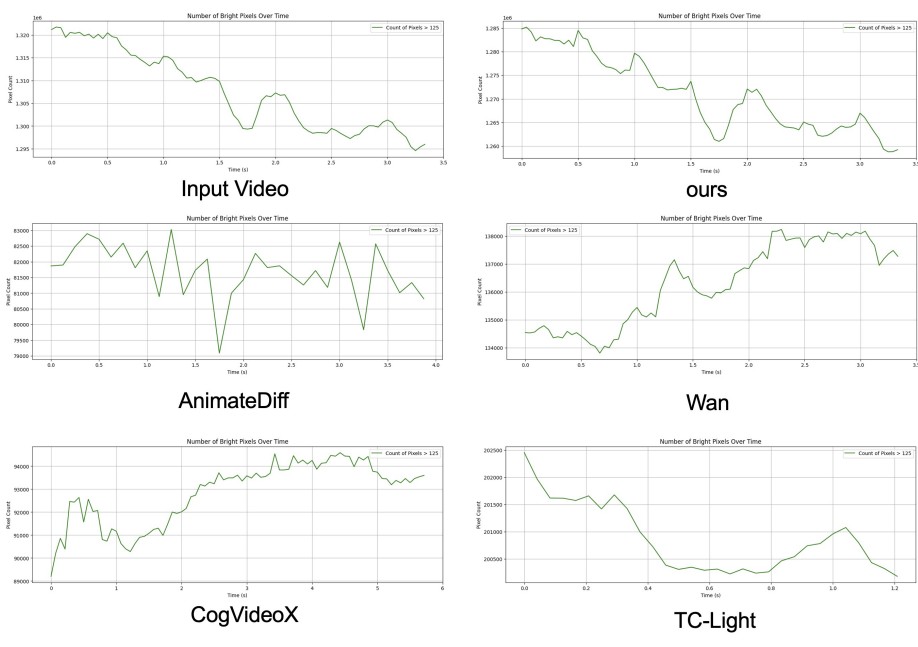

Figure 16: Plot of number of bright pixels (brightness≥125) over time.

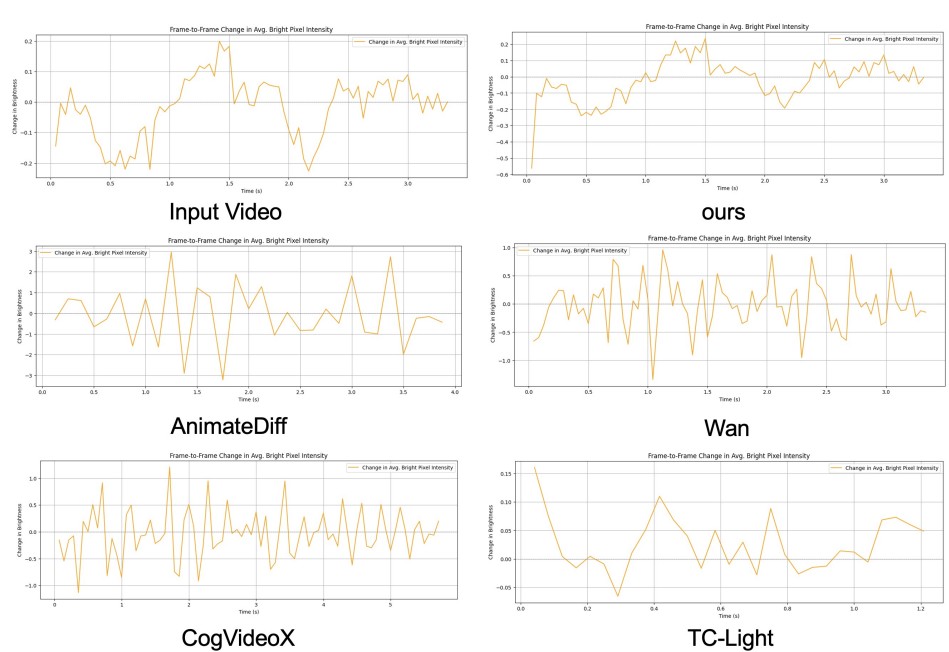

Figure 17: Plot of frame-to-frame change in average bright pixel (brightness≥125) intensity.

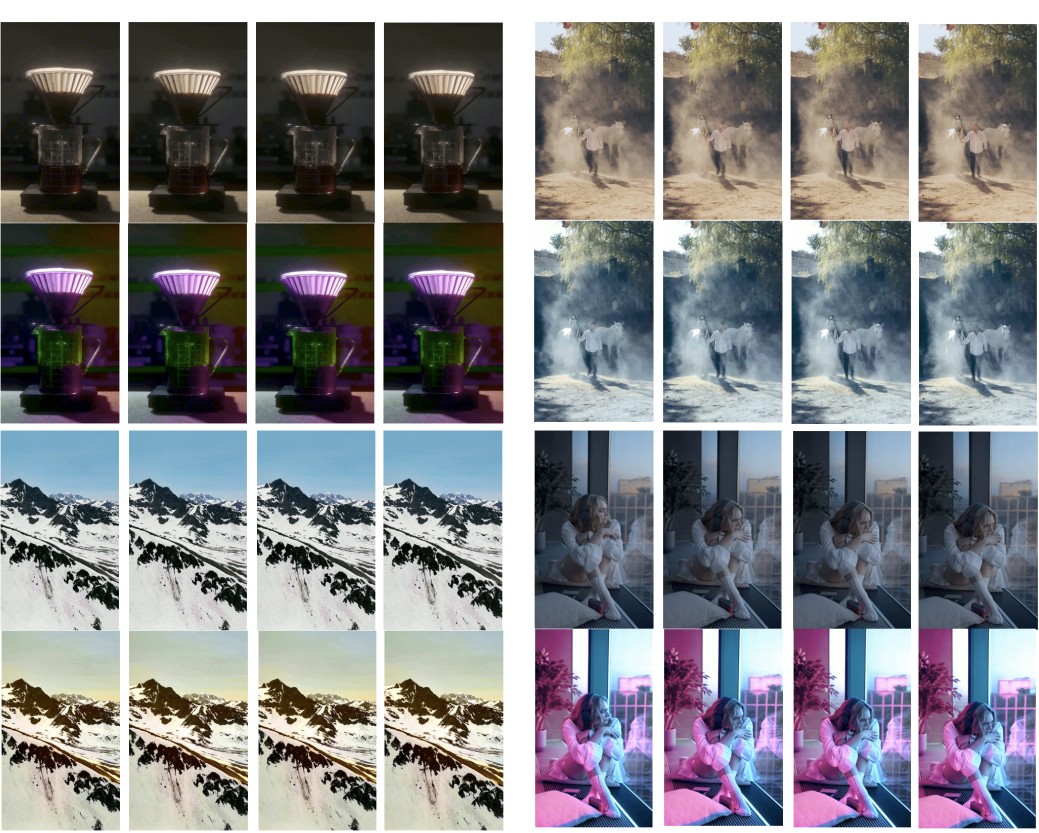

Figure 18: More Hi-Light relit video showcase.

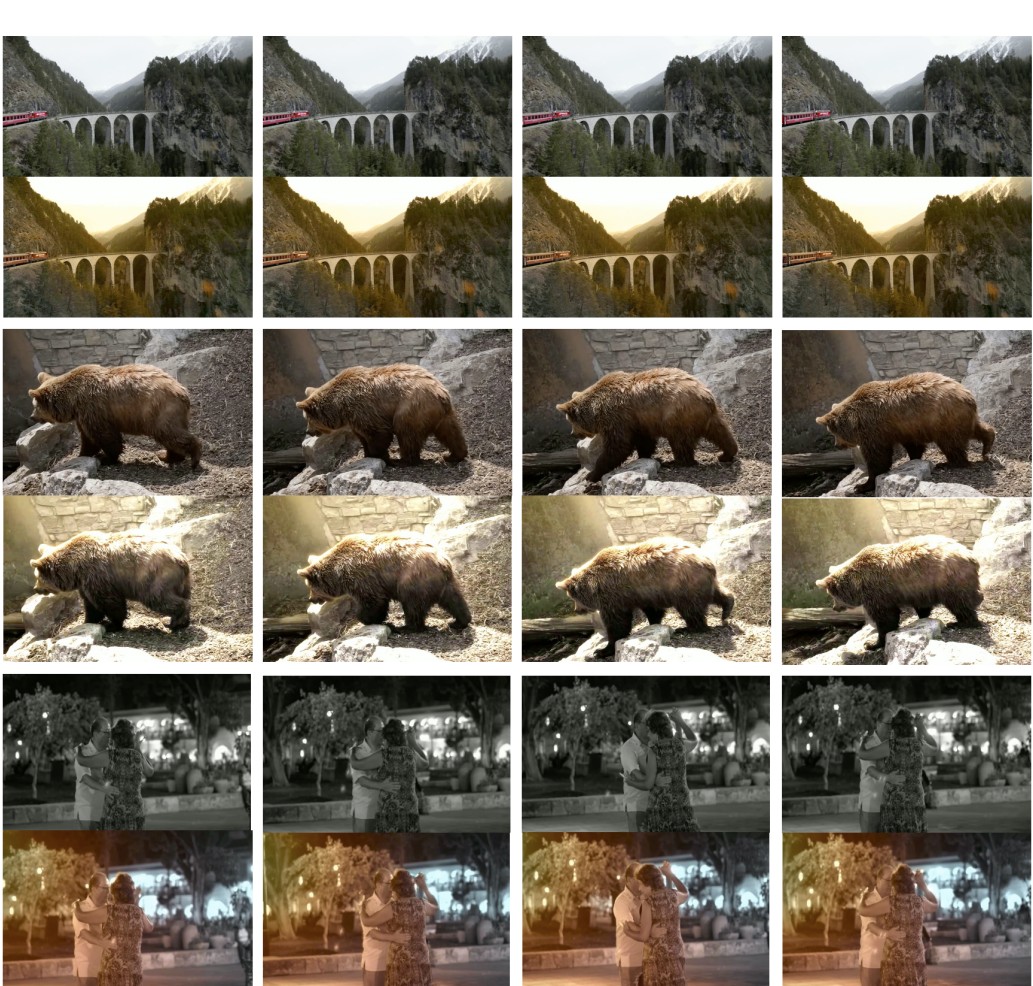

Figure 19: More Hi-Light relit video showcase.

## B  ADDITIONAL STUDIES FOR DISCUSSION

### B.1  ABLATION STUDY ON THE $\alpha$ TERM IN THE OPTICAL FLOW LIGHT SMOOTHING FILTER

The ablation results show that increasing the fixed $\alpha$ from 0.1 to 0.9 consistently improves all light-stability metrics, while SSIM remains nearly unchanged. This indicates that a large $\alpha$ provides the best trade-off for suppressing highlight flicker without visibly degrading per-frame details. However, at very high values, we occasionally observe subtle motion trails in fast-moving regions when inspected closely. In contrast, the adaptive $\alpha$ variant achieves slightly better performance across all light stability scores. By reducing the smoothing weight only in high-flow areas, the adaptive formulation follows the temporal behaviour of the fixed $\alpha$ but avoids oversmoothing motion boundaries. The adaptive scores track just below the corresponding fixed-$\alpha$, confirming that they preserve most of the temporal stability while preventing motion trails on fast-moving subjects.

| Fixed $\alpha$ | SSIM | $S_I$ ($\uparrow$) | $S_c$ ($\uparrow$) | $S_j$ ($\uparrow$) | $S_{LS}$ ($\uparrow$) |
|---|---|---|---|---|---|
| 0.1 | 0.657 | 0.349 | 0.378 | 0.500 | 0.409 |
| 0.2 | 0.657 | 0.362 | 0.386 | 0.509 | 0.419 |
| 0.3 | 0.656 | 0.374 | 0.394 | 0.513 | 0.427 |
| 0.4 | 0.656 | 0.389 | 0.405 | 0.528 | 0.441 |
| 0.5 | 0.655 | 0.401 | 0.416 | 0.541 | 0.453 |
| 0.6 | 0.655 | 0.418 | 0.428 | 0.553 | 0.466 |
| 0.7 | 0.654 | 0.436 | 0.439 | 0.561 | 0.478 |
| 0.8 | 0.653 | 0.450 | 0.453 | 0.568 | 0.490 |
| 0.9 | 0.654 | 0.460 | 0.463 | 0.574 | 0.499 |
| Adaptive $\alpha$ | | | | | |
| 0.1 | 0.657 | 0.354 | 0.381 | 0.502 | 0.412 |
| 0.3 | 0.657 | 0.374 | 0.395 | 0.510 | 0.426 |
| 0.5 | 0.656 | 0.403 | 0.419 | 0.542 | 0.455 |
| 0.7 | 0.655 | 0.438 | 0.447 | 0.556 | 0.480 |
| 0.9 | 0.653 | 0.470 | 0.467 | 0.584 | 0.507 |

Table 10: Ablation results for the fixed and adaptive $\alpha$ terms in the optical-flow light smoothing filter.

### B.2  ABLATION STUDY ON THE $\beta$ TERM IN LAB-DF

We conducted an ablation study on 60 videos to examine the effect of the merging-strength parameter $\beta$. As shown in Table 11, increasing $\beta$ leads to a consistent drop in both SSIM and $S_{LS}$, indicating reduced detail preservation and lower light stability. A greater drop occurs between $\beta = 0.4$ and $\beta = 0.7$, suggesting that excessively large merging strengths overly suppress the fine-grained information during LAB-DF. Notably, even with $\beta$ as high as 0.9, our method continues to surpass all baseline methods by a significant margin, demonstrating robust performance under extreme merging strengths.

| $\beta$ | SSIM | $S_I$ ($\uparrow$) | $S_c$ ($\uparrow$) | $S_j$ ($\uparrow$) | $S_{LS}$ ($\uparrow$) |
|---|---|---|---|---|---|
| 0.1 | 0.974 | 0.485 | 0.474 | 0.536 | 0.498 |
| 0.2 | 0.962 | 0.475 | 0.457 | 0.534 | 0.489 |
| 0.3 | 0.945 | 0.461 | 0.448 | 0.527 | 0.479 |
| 0.4 | 0.926 | 0.436 | 0.433 | 0.518 | 0.462 |
| 0.5 | 0.906 | 0.408 | 0.419 | 0.506 | 0.444 |
| 0.6 | 0.887 | 0.373 | 0.399 | 0.494 | 0.422 |
| 0.7 | 0.868 | 0.369 | 0.387 | 0.491 | 0.416 |
| 0.8 | 0.850 | 0.359 | 0.379 | 0.477 | 0.405 |
| 0.9 | 0.833 | 0.347 | 0.363 | 0.487 | 0.399 |

Table 11: Merge Strength Ablation Study results.

### B.3 ABLATION STUDY ON VIDEOS WITH DIFFERENT MOTION SPEEDS

We further evaluate the robustness of our framework under varying scene dynamics. A total of 60 videos with manually selected motion levels (equally split among fast, medium, and slow) were used in this ablation. The results in Table 12 show that motion speed has a negligible impact on detail preservation. SSIM varies by at most 0.005 across the three groups, and the change in $S_{LS}$ remains within 1%. These findings indicate that our proposed method maintains consistent performance regardless of motion intensity, demonstrating strong robustness to variations in scene dynamics.

| Motion | SSIM | $S_I$ ($\uparrow$) | $S_c$ ($\uparrow$) | $S_j$ ($\uparrow$) | $S_{LS}$ ($\uparrow$) |
|--------|------|------|------|------|------|
| Slow | 0.947 | 0.468 | 0.459 | 0.511 | 0.480 |
| Medium | 0.952 | 0.501 | 0.461 | 0.537 | 0.499 |
| Fast | 0.949 | 0.487 | 0.454 | 0.530 | 0.490 |

Table 12: Ablation on motion speed. Performance metrics across videos with slow, medium, and fast motion show minimal variation, indicating strong robustness to scene dynamics.

### B.4 COMPARATIVE EXPERIMENTAL RESULTS ON MODELS' RUNTIME

We measured the average time taken for the models to relight 60 videos, each of which has 81 frames. This set of experiments was conducted using an H100 GPU.

| Model | Scaled Average Relighting Time |
|-------|--------------------------------|
| LAV (AnimateDiff) | 276s |
| TC-Light | 398S |
| LAV (Wan) | 476s |
| LAV (CogVideoX) | 503s |
| Hi-Light (ours) | 530s |

Table 13: Runtime comparison results in ascending order. The average time taken for the models to relight videos of 81 frames.

### B.5 EVALUATION USING VBENCH AND OTHER METRICS

To further validate our model's performance and the soundness of our evaluation paradigm, we employed VBench (Huang et al., 2024), a widely adopted benchmark for video generation. Note that Hi-Light relit videos were downsampled to 1080p to satisfy VRAM constraints for the motion smoothness and dynamic degree calculations. In video relighting, the primary objective is to modify illumination while preserving the original scene's details and motion dynamics. Consequently, metrics such as Motion Smoothness, Dynamic Degree, Subject Consistency and Background Consistency should be nearly identical to those of the original video. As shown in Table 14, Hi-Light demonstrates exceptional performance, matching the original video's Motion Smoothness and Dynamic Degree almost perfectly. In contrast, competitors like TC-Light and LAV exhibit significantly increased Dynamic Degrees, indicating a possible introduction of motion artifacts. Furthermore, Hi-Light preserves Subject Consistency effectively, whereas other models degrade details in identity. Notably, LAV (AnimateDiff) achieves a counter-intuitively high Background Consistency score; we attribute this to the model producing blurred yet consistent backgrounds, which the metric misinterprets as stability.

However, VBench has distinct limitations when applied to the video relighting task rather than generation. First, the Temporal Flickering metric, which relies on pixel-wise Mean Absolute Error (MAE) for static scenes, inherently misinterprets legitimate motion in high-resolution videos as artifacts. Consequently, lower-fidelity models with blurry edges receive artificially high scores because their lack of detail minimizes pixel differences, while the sharp, high-resolution motion in original and Hi-Light videos is penalized. Second, the Aesthetic Quality metric is biased toward conventionally "pretty" (bright and saturated) images, and may unfairly penalize physically accurate but

intentionally dramatic or dark lighting. Therefore, while VBench serves as a useful supplementary benchmark, our specific evaluation paradigm is better tailored to assess detail degradation and lighting stability in relighting tasks.

| | Original Video | Hi-Light (ours) | LAV (Animatediff) | LAV (CogVideoX) | LAV (Wan) | TC-Light |
|---|---|---|---|---|---|---|
| Dynamic Degree (↑) | 0.626 | 0.631 | 0.625 | 0.780 | 0.614 | 0.829 |
| Motion Smoothness (↑) | 0.985 | 0.984 | 0.982 | 0.983 | 0.987 | 0.982 |
| Temporal Flickering (↑) | 0.972 | 0.968 | 0.975 | 0.979 | 0.978 | 0.970 |
| Temporal Style (↑) | 0.192 | 0.209 | 0.197 | 0.192 | 0.201 | 0.205 |
| Subject Consistency (↑) | 0.939 | 0.937 | 0.917 | 0.933 | 0.935 | 0.911 |
| Background Consistency (↑) | 0.953 | 0.943 | 0.948 | 0.929 | 0.939 | 0.931 |
| Aesthetic Quality (↑) | 0.528 | 0.553 | 0.537 | 0.528 | 0.557 | 0.560 |
| Imaging Quality (↑) | 0.689 | 0.684 | 0.527 | 0.563 | 0.611 | 0.557 |
| FID (↓) | 0 | 76 | 241 | 133 | 135 | 120 |

Table 14: VBench and metrics from prior works.

In addition to the Motion Smoothness and CLIP scores computed in the SOTA work , we further investigated the Fréchet inception distance (FID) (Heusel et al., 2017). FID is a metric used to evaluate the quality of generated models by comparing the distribution of generated images to the distribution of real images. With reference to the evaluation of FID in Light-A-Video Zhou et al. (2025), we have done a similar evaluation as shown in Table 14.

