# OpenReview forum: "Hi-Light: A Path to high-fidelity, high-resolution video relighting with a Novel Evaluation Paradigm"
_ICLR.cc/2026/Conference — Submitted to ICLR 2026_

### Official Review · Reviewer_afb4 · 2025-10-26

**Soundness:** 2
**Presentation:** 2
**Contribution:** 2
**Rating:** 2
**Confidence:** 5

**Summary:**

This paper presents Hi-Light, a training-free framework for high-fidelity and high-resolution video relighting.
It introduces three modules: lightness-prior anchored guided diffusion for luminance stability, Hybrid Motion-Adaptive Lighting Smoothing Filter for flicker reduction, and LAB-based Detail Fusion for detail recovery, along with a new metric Light Stability Score (SLS) for evaluating temporal lighting consistency.
Experiments show that Hi-Light achieves significantly better stability and detail preservation than prior methods such as Light-A-Video and TC-Light on the proposed metric.

**Strengths:**

1) Clear motivation: addresses flickering and detail loss in video relighting, both key unsolved problems.
2) Extensive experiments and ablations demonstrate strong performance gains over prior methods in the proposed benchmark.
3) Easy to follow, well-structured, and reproducible with provided code and detailed analysis.

**Weaknesses:**

**Major Weaknesses**

1) The evaluation metric design is biased. As stated in L414–415 (“It is notably close to the original input video, which represents the ideal target for these metrics”), the chosen SSIM and consistency metrics mainly favor temporal coherence and similarity to the input video rather than the quality of relighting. This is a fundamental flaw for a video relighting paper.

2) The LAB fusion may compromise relighting quality. As shown in Fig. 8 (top row), the “LAB-DF Fused video” appears worse than the smoothed and intermediate relit videos. Directly fusing the L channel mainly preserves global tone but loses illumination contrast and shadow variation, thereby reducing the realism of relighting.

3) The proposed metric lacks novelty and is not fully justified. SSIM is widely used, and the consistency score, obtained by thresholding grayscale frames and measuring average brightness and bright pixel counts, does not effectively capture temporal consistency. A comparison with existing metrics (e.g., consistency and smoothness in VBench [1]) is necessary to clarify its advantage.

4) The HMA-LSF module relies on optical flow warping, which can be noisy. Although filtering alleviates some artifacts, large-motion cases where optical flow fails should be analyzed and visualized to assess robustness.

5) Only one visual comparison (Fig. 5) is shown, and no video comparison results are provided. In Fig. 1 (second row), the relighting quality of the woman’s right hand in the last frame is also unsatisfactory. For a video relighting or editing task, more qualitative and video-based results are essential to convincingly demonstrate performance. More qualitative and dynamic visual examples are needed to support the claims. In addition, all current examples appear to use static-camera video inputs; results under larger camera motion should also be presented to evaluate robustness.

**Minor Weaknesses**

6) The method contains many hyperparameters, and it is unclear whether they are fixed globally or tuned per scene. Robustness across diverse videos should be further analyzed.

7) Runtime comparisons with other methods are missing.

8) The paper lacks discussion on limitations and potential future work.

9) Figure presentation issues: Fig. 3 includes unexplained icons, and its caption could be more detailed. In Fig. 5, method names could be properly aligned.

**Reference**

[1] VBench: Comprehensive Benchmark Suite for Video Generative Models, CVPR 2024

**Questions:**

1) Could the authors provide relighting quality metrics, such as those used in Light-A-Video [1] and other existing consistency metrics like VBench [2], to better evaluate lighting realism and temporal stability?
2) Can more visual comparisons with baseline methods be added, including qualitative and video-based examples, to clearly demonstrate improvements in video relighting?
3) Could the authors include runtime comparisons with baseline methods?

**Reference**

[1] Light-A-Video: Training-free Video Relighting via Progressive Light Fusion, ICCV 2025

[2] VBench: Comprehensive Benchmark Suite for Video Generative Models, CVPR 2024

---

> ### Author Response · Authors · 2025-11-22
> **Discussion with Reviewer afb4 - Part 1**
>
> We sincerely appreciate the reviewer's time and constructive comments on our paper. Please see our responses below.
>
> **W1: The evaluation metric design is biased. As stated in L414–415 (“It is notably close to the original input video, which represents the ideal target for these metrics”), the chosen SSIM and consistency metrics mainly favor temporal coherence and similarity to the input video rather than the quality of relighting. This is a fundamental flaw for a video relighting paper.**
>
> We respectfully disagree that the metrics are biased or flawed. Our understanding of the  **essential prerequisites** for any successful video editing task: (1) applying new stable and realistic lighting as integrity, and (2) preserving the realistic and intrinsic content of the input videos as fidelity.
>
> The artifacts introduced by diffusion models, specifically detail degradation and temporal flickering, are objective failures that our metrics quantify. When we describe the input video as the "ideal target," we refer specifically to its structural content and temporal coherence, not its lighting aesthetic quality. A valid relighting method should project illumination without jeopardizing the underlying geometry or subject identity. Therefore, our use of SSIM and the Light Stability Score **does not penalize relighting effects**, but rather **penalizes the destructive artifacts** that exist in current baselines [1, 2], ensuring that the method operates as a controllable edit rather than a destructive style transfer.
>
>
>
> **W2: The LAB fusion may compromise relighting quality. As shown in Fig. 8 (top row), the “LAB-DF Fused video” appears worse than the smoothed and intermediate relit videos. Directly fusing the L channel mainly preserves global tone but loses illumination contrast and shadow variation, thereby reducing the realism of relighting.**
>
> We acknowledge that figures cannot fully capture the dynamic nuances of the relit videos, and we agree that our LAB-DF module represents a design trade-off that prioritizes structural realism over raw lighting intensity. To accommodate varying aesthetic preferences, our released code includes post-processing tools allowing users to fine-tune saturation and brightness.
>
> Regarding the presentation in Figure 8, we apologize for its limitations in illustrating these comparative advantages. **We have uploaded comprehensive new comparison videos to our anonymous GitHub repository**, which we believe better demonstrate the superior fidelity and stability of our approach.
>
>
> **W3: The proposed metric lacks novelty and is not fully justified. SSIM is widely used, and the consistency score, obtained by thresholding grayscale frames and measuring average brightness and bright pixel counts, does not effectively capture temporal consistency. A comparison with existing metrics (e.g., consistency and smoothness in VBench [1]) is necessary to clarify its advantage.**
>
> We wish to clarify that we do not claim the SSIM metric itself as a novel contribution. Rather, our contribution is the establishment of the first principled evaluation paradigm tailored for video relighting, a task that does not have a standardized assessment protocol. Previous works (e.g., Light-A-Video [1], TC-Light [2]) rely on generative metrics like Fréchet Inception Distance (FID) or CLIP scores. While useful for general image generation, these metrics fail to capture the detail degradation and light flickering problems. We deliberately chose SSIM because video relighting is a reconstruction and enhancement task, and SSIM does not need a GPU to compute.
>
> By adopting the reviewer’s suggestion, we have incorporated additional evaluations using VBench and established metrics from prior work [1, 2] in Appendix B.5. These new results actually provide a strong justification for our evaluation paradigm.
>
>
> **W4: The HMA-LSF module relies on optical flow warping, which can be noisy. Although filtering alleviates some artifacts, large-motion cases where optical flow fails should be analyzed and visualized to assess robustness.**
>
> We agree with the reviewer and have **newly conducted an ablation study on the robustness of the light smoothing filter.** The new results are included in Appendix B.1, please have a look.

---

> > ### Author Response · Authors · 2025-11-22
> > **Discussion with Reviewer afb4 - Part 2**
> >
> > **W5: Only one visual comparison (Fig. 5) is shown, and no video comparison results are provided. In Fig. 1 (second row), the relighting quality of the woman’s right hand in the last frame is also unsatisfactory. For a video relighting or editing task, more qualitative and video-based results are essential to convincingly demonstrate performance. More qualitative and dynamic visual examples are needed to support the claims. In addition, all current examples appear to use static-camera video inputs; results under larger camera motion should also be presented to evaluate robustness.**
> >
> > We fully acknowledge that the initial lack of sufficient demo videos and the resolution limitations of Figure 1 were oversights in our presentation. We have uploaded a comprehensive set of comparison videos to our anonymous GitHub repository. Moreover, this update includes specific demonstrations featuring large camera motions to rigorously verify robustness in dynamic scenarios. We are confident that these new visual materials provide convincing evidence of our method's superior fidelity and stability.
> >
> >
> > **W6: The method contains many hyperparameters, and it is unclear whether they are fixed globally or tuned per scene. Robustness across diverse videos should be further analyzed.**
> >
> > We wish to clarify that all hyperparameters in our framework are **globally fixed** across all experiments reported in the paper. We do not tune parameters on a per-scene basis, which highlights the generalization capability of our method. We updated the information in the new version.
> >
> > To further address the concern regarding robustness across diverse videos, we have included a **new robustness study on varying motion speeds in Appendix B.3**. In short, the results suggest our method is robust to the change in the motion speed.
> >
> >
> >
> > **W7: Runtime comparisons with other methods are missing.**
> >
> > We thank the reviewer for this suggestion. Although optimizing for speed was not the primary objective of this work, which focuses on high-fidelity and temporal stability.
> >
> > We agree that computational benchmarks are valuable for the community. We have added a runtime comparison against baseline methods in Appendix B.4. These results demonstrate the practical efficiency of our approach even with the additional stability and detail-preservation modules.
> >
> >
> > **W8: The paper lacks discussion on limitations and potential future work.**
> >
> > **Due to the page limit, we did not include the Limitations and Future Work section. We have added it to the new updated version. Please refer to the newly updated paper.**
> >
> >
> > **W9: Figure presentation issues: Fig. 3 includes unexplained icons, and its caption could be more detailed. In Fig. 5, method names could be properly aligned.**
> >
> > We appreciate the reviewer’s attention to detail regarding the figures. We have updated the manuscript.
> > Figure 3: We have expanded the caption to provide a more detailed explanation of the pipeline.
> > Figure 5: We have corrected the layout to ensure that all method names are properly aligned and consistent.

---

> ### Author Response · Authors · 2025-11-22
> **Discussion with Reviewer afb4 - Part 3**
>
> **Q1:Could the authors provide relighting quality metrics, such as those used in Light-A-Video [1] and other existing consistency metrics like VBench [2], to better evaluate lighting realism and temporal stability?**
>
> We thank the reviewer for pointing out this omission. To ensure a comprehensive comparison, we have **updated Appendix B.5 to include evaluations using standard metrics from prior work [1, 2]**, including FID, CLIP scores (included in VBench), and the VBench results.
>
> While we provide these metrics for completeness, we respectfully argue that they are fundamentally ill-suited for the specific task of video relighting. Metrics like FID and CLIP measure distributional quality and semantic alignment, respectively and are suited for image generation. These metrics including VBench fail to penalize the two most critical artifacts in this domain: detail degradation and light flickering.
>
> This highlights the significance of our proposed evaluation paradigm tailored for video relighting. Unlike heavy deep-learning metrics, our approach is computationally efficient, interpretable, and physically grounded. It explicitly targets the preservation of structural integrity and temporal consistency, providing a far more accurate diagnostic tool for the relighting task than generic generative benchmarks.
>
> **Q2:Can more visual comparisons with baseline methods be added, including qualitative and video-based examples, to clearly demonstrate improvements in video relighting?**
>
> We are sorry for missing the video comparisons. We have uploaded extensive videos to the **anonymous GitHub link** to show the significant improvements. Please have a look.
>
>
> **Q3:Could the authors include runtime comparisons with baseline methods?**
>
> We have added a runtime comparison to the updated paper in Appendix B.4. Please refer to the new uploaded version.
>
> **List of changes for the updated paper:**
> 1. Included ablation studies on alpha in Equation (7) and beta in Equation (9), as well as evaluations on videos with different motion speeds.
>
> 2. Evaluated the models using VBench and metrics from prior works.
>
> 3. Uploaded an extensive set of demo videos to the anonymous GitHub page for comparison and ablation studies.
>
> 4. Added a "Limitations and Future Work" section.
>
> 5. Refined figure captions for greater detail.
>
> **Highlights of our scope and contributions:**
>
> 1. We tackle the critical challenges of detail loss and flickering prevalent in SOTA approaches [1, 2].
>
> 2. Our method stands as the only current solution for high-resolution video relighting.
>
> 3. We introduce a new evaluation paradigm explicitly designed for this task, empirically demonstrating that it provides a more accurate and robust assessment of relighting quality.
>
> **References:**
>
> [1] Light-A-Video: Training-free Video Relighting via Progressive Light Fusion. (ICCV 2025)
>
> [2] TC-Light: Temporally Coherent Generative Rendering for Realistic World Transfer (NeurIPS 2025)

---

> > ### Comment · Reviewer_afb4 · 2025-11-27
> >
> > I acknowledge the authors' efforts in addressing some of the initial concerns during the rebuttal period, including providing additional experiments, visualizations, and video results. The manuscript has undoubtedly improved from its initial version. I appreciate the authors' diligent work and detailed responses.
> >
> > However, my primary concern remains fundamentally unaddressed. For a paper on video relighting, the core evaluation paradigm, positioned as a key contribution, focuses solely on temporal consistency and similarity to the input video (via SSIM and the proposed SLS). It does not adequately evaluate the quality of the relighting itself, such as the realism, aesthetic appeal, and physical plausibility of the new illumination. This creates a critical gap, as a method could, in theory, perform well on these metrics by making minimal changes to the original lighting, which is not the end goal of the task.
> >
> > While the authors argue that preserving input fidelity is a prerequisite, a convincing relighting framework must also be judged on its ability to apply new, high-quality lighting effects. The inclusion of metrics from VBench and prior works in the appendix is a step, but it does not resolve the core issue that the proposed paradigm itself is incomplete and potentially biased.
> >
> > After reviewing the authors' response and considering the other reviewers' comments, I find that this fundamental shortcoming affects both the evaluation of their proposed method and the claimed contribution of their new evaluation paradigm. In my view, this dual limitation still places the paper below the acceptance bar for ICLR. For future work, I strongly recommend the authors develop and incorporate metrics that directly assess relighting quality in both their method evaluation and evaluation paradigm  design to provide a more balanced and meaningful assessment framework.

---

> > > ### Author Response · Authors · 2025-11-27
> > > **Discussion with Reviewer afb4 - Part 4**
> > >
> > > We fully respect the reviewer’s decision. While we value the feedback, we would like to clarify the **scope and contributions** of our work regarding the evaluation concerns:
> > >
> > > 1. **Feasibility of Aesthetic and Physics Metrics**
> > >
> > > We would like to emphasize that our evaluation paradigm **effectively targets the notorious problems in video relighting**. We agree that measuring physical plausibility and aesthetic appeal is the ultimate goal for video generation. **However, accurately quantifying these aspects remains an open research question across the entire domain**, with no currently established automated standards. In the absence of such metrics, we relied on the gold standard for these subjective qualities: a blind professional human study (Appendix), where respondents consistently preferred our results and an established benchmark, VBench.
> > >
> > > 2. **Robustness of Evaluation**
> > >
> > > We respectfully disagree that the evaluation is biased. **To ensure fairness, we did not rely on our metrics alone; we provided extensive ablation studies, open-source code, qualitative comparisons, and a blind human study.**
> > >
> > > 3. **Independent Contribution of the Method**
> > >
> > > Crucially, our contribution is not limited to the evaluation paradigm. Even setting aside our proposed metrics, our method **significantly outperforms SOTA baselines [1, 2] on established metrics and in a human study**. This demonstrates that our **methodological contribution stands independently as a significant advancement** in the field. Therefore, the proposed evaluation paradigm serves as a **valuable enhancement to an already robust methodological contribution**, rather than being the primary basis of our contributions.
> > >
> > > **References:**
> > >
> > > [1] Light-A-Video: Training-free Video Relighting via Progressive Light Fusion. (ICCV 2025)
> > >
> > > [2] TC-Light: Temporally Coherent Generative Rendering for Realistic World Transfer (NeurIPS 2025)

---

### Official Review · Reviewer_C9X2 · 2025-10-28

**Soundness:** 2
**Presentation:** 1
**Contribution:** 2
**Rating:** 2
**Confidence:** 4

**Summary:**

The paper presents a training-free, high-resolution video relighting framework designed to generate relit videos without degrading high-frequency details from the input and while maintaining better temporal stability. Specifically, the paper introduces a lighting prior as auxiliary information to reduce luminance fluctuations at low resolution, a hybrid motion-adaptive light smoothing filter to mitigate flicker, and a LAB-preservation fusion strategy to recover high-frequency details. Additionally, the paper proposes an evaluation metric to measure temporal lighting stability.

**Strengths:**

- The presented relighting method appears to be a plug-and-play framework that can integrate seamlessly with various video diffusion models, making it highly practical for real-world applications.
- The proposed approach supports high-resolution video relighting, which significantly enhances its utility for real-world use cases.

**Weaknesses:**

- The paper suggests that prior methods degrade high-frequency details such as hair or foliage; however, it does not provide sufficient qualitative evidence to demonstrate that the proposed approach retains these details more effectively in relit videos.
- The paper introduces a new lighting stability score along with an adapted SSIM metric. However, it does not include evaluations using existing metrics from prior work, which would enable a fairer comparison with baselines.
- The paper claims improved computational efficiency by performing relighting at low resolution. This claim would be stronger if runtime comparisons with baseline methods were provided to demonstrate computational effectiveness.
- The paper proposes leveraging optical flow to reduce temporal lighting flicker but does not provide details on how the optical flow is generated or evaluated for robustness.
- The paper explains the combination of low-resolution lightness with high-frequency details but does not clarify how LAB components are upscaled or how artifacts, if any, such as color bleeding, desaturation, and ghosting are mitigated.
- The paper does not address how hard shadows or highlights are handled when relighting under different lighting directions or sources. The presented qualitative results do not capture this scenario.
- Are there cases where the proposed pipeline fails to generate a satisfactory relit video? Including failure cases would provide a more comprehensive evaluation of the method’s limitations.
- Figure 3 is difficult to interpret, and the caption lacks sufficient detail to explain the framework. Adding relevant labels and descriptive terms to the figure would make the pipeline easier to understand.

**Questions:**

- The proposed metric primarily measures temporal lighting stability. How do you ensure color consistency over time? Why not evaluate with existing metrics such as VBench [a] for temporal flicker, motion smoothness, aesthetic quality, and background consistency?
- The paper states that \alpha in Eq. 7 adapts to motion magnitude but provides no qualitative analysis of its impact.

Additional references:

[a] Vbench: Comprehensive benchmark suite for video generative models. CVPR 2024.

---

> ### Author Response · Authors · 2025-11-22
> **Discussion with Reviewer C9X2 - Part 1**
>
> We sincerely thank the reviewer for their insightful comments and the time spent evaluating our paper. Please see our responses below.
>
> **W1: The paper suggests that prior methods degrade high-frequency details such as hair or foliage; however, it does not provide sufficient qualitative evidence to demonstrate that the proposed approach retains these details more effectively in relit videos.**
>
> We apologize for insufficient qualitative evidence. We have uploaded **new, comprehensive video demos** to the anonymous link. Please have a look.
>
> **W2: The paper introduces a new lighting stability score along with an adapted SSIM metric. However, it does not include evaluations using existing metrics from prior work, which would enable a fairer comparison with baselines.**
>
> We thank the reviewer for pointing out this omission. To ensure a comprehensive comparison, we have updated Appendix B.5 to include evaluations using standard metrics from prior work, including FID, CLIP scores (included in VBench), and the VBench results.
>
> While we provide these metrics for completeness, we respectfully argue that they are fundamentally ill-suited for the specific task of video relighting. Metrics like FID and CLIP measure distributional quality and semantic alignment, respectively and are suited for image generation.  These metrics fail to penalize the two most critical artifacts in this domain: detail degradation and light flickering.
>
> This highlights the significance of our proposed evaluation paradigm tailored for video relighting. Unlike heavy deep-learning metrics, our approach is computationally efficient, interpretable, and physically grounded. It explicitly targets the preservation of structural integrity and temporal consistency, providing a far more accurate diagnostic tool for the relighting task than generic generative benchmarks.
>
>
> **W3: The paper claims improved computational efficiency by performing relighting at low resolution. This claim would be stronger if runtime comparisons with baseline methods were provided to demonstrate computational effectiveness.**
>
> We apologize for any confusion regarding our motivation for downsampling. We wish to clarify that we do not claim downsampling as an accelerator for model inference. To clarify the claims originally in Lines 225-226 (we have clarified them in the updated version), this is a **necessary constraint strategy** for the following reasons:
>
> 1. Downsampling to 480p is required to fit the memory constraints of standard consumer GPUs during the diffusion process.
>
> 2. Current open-source video diffusion backbones are typically trained on resolutions 480p and 720p. Hence, the diffusion models have restrictions on the input video.
>
> 3. Our contribution is not the downsampling itself, but the resolution-independent architecture (LAB-DF) that allows us to take this low-res 480p relit output and successfully transfer the illumination information to high-res videos, bypassing the need to run the heavy diffusion model at native high resolution, which is not feasible now.
>
> To address the reviewer's request for empirical data, we have included a runtime comparison in Appendix B.4. This demonstrates the practical trade-offs and confirms that our pipeline remains competitive even with the additional restoration steps. Nevertheless, runtime is not prioritized, given that the quality of relighting is not guaranteed in the existing work [1, 2].
>
>
> **W4: The paper proposes leveraging optical flow to reduce temporal lighting flicker but does not provide details on how the optical flow is generated or evaluated for robustness.**
>
> Instead of utilizing RAFT, a popular transformer-based optical flow model that imposes high GPU computational costs for video processing. Therefore, we obtained the Farneback optical flow using the OpenCV library, which only uses CPU for better accessibility.
> We newly included an ablation study on the optical flow filter in Appendix B.1 to show the robustness of our design.
>
>
> **W5: The paper explains the combination of low-resolution lightness with high-frequency details but does not clarify how LAB components are upscaled or how artifacts, if any, such as color bleeding, desaturation, and ghosting are mitigated.**
>
> We apologize for the confusion caused. The smoothed intermediate relit video will be upsampled back to its original input size and then extract the illumination information.
>
> It is a very insightful question. We indeed encountered desaturation when using the weighted sum blending in LAB-DF. We have two solutions which are available in our demo scripts. First, we offer an alternative blending mode that up-scales the illumination information of the raw video based on the extracted illumination information. Second, we offer a saturation scaling function to scale up the saturation of the videos.  The ghosting problem is effectively eliminated by HMA-LSF and LAB-DF. Please refer to the newly uploaded videos in our anonymous GitHub link.

---

> ### Author Response · Authors · 2025-11-22
> **Discussion with Reviewer C9X2 - Part 2**
>
> **W6: The paper does not address how hard shadows or highlights are handled when relighting under different lighting directions or sources. The presented qualitative results do not capture this scenario.**
>
> This is a professional observation related to lighting. We openly acknowledge that precise control over these physical phenomena is a shared challenge across the entire domain of text-prompted video relighting, including our work and recent SOTA baselines like Light-A-Video [1] and TC-Light [2].  It is commonly done through physics-based 3D geometry rendering for 3D views of images (e.g. DiffusionRenderer [3]).
>
> Furthermore,  objectively evaluating shadows and highlights in text-controlled video relighting is currently infeasible. While our current work solves the critical issues of temporal flickering and detail degradation, we agree that integrating geometric constraints to enable fine-grained shadow control is a complex but valuable direction for future research. We have updated our Limitations and Future Work section to discuss this challenge.
>
> **W7: Are there cases where the proposed pipeline fails to generate a satisfactory relit video? Including failure cases would provide a more comprehensive evaluation of the method’s limitations.**
>
> Honestly, we did not encounter any big failure, but we do have some unsatisfactory relit videos. If the input video is already exposed to strong light, the relit video will have a smaller relighting effect on the highlight part, which is also a problem for the baselines. We will update this to the Limitations and Future Work section.
>
> **W8: Figure 3 is difficult to interpret, and the caption lacks sufficient detail to explain the framework. Adding relevant labels and descriptive terms to the figure would make the pipeline easier to understand.**
>
> We have improved the caption in the updated version for better understanding.
>
>
> **Q1: The proposed metric primarily measures temporal lighting stability. How do you ensure color consistency over time? Why not evaluate with existing metrics such as VBench [a] for temporal flicker, motion smoothness, aesthetic quality, and background consistency?**
>
> Our work same as the published SOTA models [1, 2] focus on static lighting, and color inconsistency is not observed. We anticipate it to occur in future work on dynamic lighting.
>
> It is an excellent idea to include VBench, and we have newly added it to the updated version in Appendix B.5. The results suggest that our evaluation paradigm is better suited for video relighting. Please refer to the updated paper for details.
>
>
> **Q2: The paper states that \alpha in Eq. 7 adapts to motion magnitude but provides no qualitative analysis of its impact.**
>
> We have newly included a qualitative analysis of the impact of alpha for both fix and adaptive status in Appendix B.1.
>
>
>
>
> **List of changes for the updated paper:**
> 1. Included ablation studies on alpha in Equation (7) and beta in Equation (9), as well as evaluations on videos with different motion speeds.
>
> 2. Evaluated the models using VBench and metrics from prior works.
>
> 3. Uploaded an extensive set of demo videos to the anonymous GitHub page for comparison and ablation studies.
>
> 4. Added a "Limitations and Future Work" section.
>
> 5. Refined figure captions for greater detail.
>
> **Highlights of our scope and contributions:**
>
> 1. We tackle the critical challenges of detail loss and flickering prevalent in SOTA approaches [1, 2].
>
> 2. Our method stands as the only current solution for high-resolution video relighting.
>
> 3. We introduce a new evaluation paradigm explicitly designed for this task, empirically demonstrating that it provides a more accurate and robust assessment of relighting quality.
>
>
> **References**:
>
> [1] Light-A-Video: Training-free Video Relighting via Progressive Light Fusion. (ICCV 2025)
>
> [2] TC-Light: Temporally Coherent Generative Rendering for Realistic World Transfer (NeurIPS 2025)
>
> [3] DiffusionRenderer: Neural Inverse and Forward Rendering with Video Diffusion Models (CVPR 2025)

---

### Official Review · Reviewer_AhoR · 2025-10-29

**Soundness:** 3
**Presentation:** 3
**Contribution:** 3
**Rating:** 6
**Confidence:** 4

**Summary:**

The paper presents Hi-Light, a training-free, backbone-agnostic workflow for video relighting, motivated by observed limitations in existing methods: temporal lighting flicker, detail degradation in high-resolution content, and the absence of a metric dedicated to lighting stability. With the designed pipeline, the method uses multiple off-the-shelf video diffusion backbones to improve temporal stability and visual details. This method formalizes an evaluation setup and introduces the Light Stability Score (SLS) to quantify temporal smoothness alongside fidelity metrics such as SSIM, aiming to provide a more complete assessment of relighting quality. Empirical results indicate state-of-the-art performance, with reported SSIM of 0.943 and SLS of 0.509, and comparative analyses showing improvements over prior baselines; human studies are reported to correlate with these metrics, supporting the validity of the evaluation.

**Strengths:**

The method leverages properties of the color space to improve video stability. It requires no additional training and can be implemented with existing video diffusion backbones. The idea is elegant and, according to experimental results, effective.

The proposed new metric makes comparisons between methods systematic.

**Weaknesses:**

In the demo videos presented by the authors, there are no scenes with significant motion, and the experiments do not include a separate analysis or comparison of scene motion. Since the algorithm relies on optical flow, it is likely to be sensitive to motion. Therefore, it remains unclear whether the method can be applied to videos with larger or more complex motion.

Although the authors claim this is a relighting task, the control over the light sources is neither very fine-grained nor accurate. The authors mention that it  support controll of the direction of light. However, the corresponding figures and references cannot be found in the paper, and the existing figures also fail to effectively demonstrate the effect of controlling the light source direction. The current results appear more like a form of video stylization that emphasizes lighting differences.

**Questions:**

How much motion can the proposed method handle in the video?
During relighting, how can the direction of the light source be controlled? If it can be controlled, how fine is the level of control?

---

> ### Author Response · Authors · 2025-11-22
> **Discussion with Reviewer AhoR - Part 1**
>
> We sincerely thank the reviewer for their constructive comments and the time spent evaluating our paper. Please see our responses below.
>
> **W1: In the demo videos presented by the authors, there are no scenes with significant motion, and the experiments do not include a separate analysis or comparison of scene motion. Since the algorithm relies on optical flow, it is likely to be sensitive to motion. Therefore, it remains unclear whether the method can be applied to videos with larger or more complex motion.**
>
> To address this, we have added **an ablation study on motion speed in Appendix B.3**, and included new demo videos in our **anonymous GitHub repository**. The performance metrics remain remarkably consistent regardless of motion speed. The minor variations observed, where 'Fast' motion marginally outperforms 'Slow', likely stem from the inherent stochasticity of the model or intrinsic differences across the input videos, rather than a systematic sensitivity to motion speed.
>
> **W2: Although the authors claim this is a relighting task, the control over the light sources is neither very fine-grained nor accurate. The authors mention that it support controll of the direction of light. However, the corresponding figures and references cannot be found in the paper, and the existing figures also fail to effectively demonstrate the effect of controlling the light source direction. The current results appear more like a form of video stylization that emphasizes lighting differences.**
>
> Regarding the concerns about fine-grained control and the distinction between relighting and stylization, we wish to clarify that our work operates within the paradigm of text-driven generative video relighting, different from physics-based rendering methods like DiffusionRenderer that require static 3D geometry. Our objective is like the SOTA  methods, such as Light-A-Video [1] and TC-Light [2], to enable light control for in-the-wild videos where such 3D data is unavailable.
>
> We apologize that static images in the paper failed to demonstrate our work. We have uploaded new demo videos to our anonymous repository showing light-controlled and visual comparison among the baselines. We encourage the reviewer to check out the updated repository.
>
>
> **Q1: How much motion can the proposed method handle in the video? During relighting, how can the direction of the light source be controlled? If it can be controlled, how fine is the level of control?**
>
> We have demo videos for high-speed motion in the repository and a respective ablation study in Appendix B.3.
>
> Due to the page limit, we included the direction control in the Appendix (Line 609-610) in the previous version and now in (Line 377). It can be controlled from four directions using text prompts: left, right, bottom, and top. A finer control of light direction can be a future work that requires fine-tuning the model. The scope of the current work is to tackle the detail degradation and light flickering problems.
>
>
> **List of changes for the updated paper:**
> 1. Included ablation studies on alpha in Equation (7) and beta in Equation (9), as well as evaluations on videos with different motion speeds.
>
> 2. Evaluated the models using VBench and metrics from prior works.
>
> 3. Uploaded an extensive set of demo videos to the anonymous GitHub page for comparison and ablation studies.
>
> 4. Added a "Limitations and Future Work" section.
>
> 5. Refined figure captions for greater detail.
>
> **Highlights of our scope and contributions:**
>
> 1. We tackle the critical challenges of detail loss and flickering prevalent in SOTA approaches [1, 2].
>
> 2. Our method stands as the only current solution for high-resolution video relighting.
>
> 3. We introduce a new evaluation paradigm explicitly designed for this task, empirically demonstrating that it provides a more accurate and robust assessment of relighting quality.
>
> **References:**
>
> [1] Light-A-Video: Training-free Video Relighting via Progressive Light Fusion. (ICCV 2025)
>
> [2] TC-Light: Temporally Coherent Generative Rendering for Realistic World Transfer (NeurIPS 2025)

---

### Official Review · Reviewer_frAN · 2025-11-02

**Soundness:** 2
**Presentation:** 3
**Contribution:** 1
**Rating:** 2
**Confidence:** 2

**Summary:**

The paper proposes a new video relighting framework called "Hi-Light" composed of three key components: i) a lightness-prior-based fusion scheme that mitigates luminance oscillations via diffusion; ii) two Plug-and-Play (PnP) filters called Hybrid Motion Adaptive Light Smoothness Prior (HMA-LSF) and LAB Detail-Preserving Fusion (LAB-DF) that remove flickers and restore texture respectively, and iii) a new quantitative metric call Light Stability Score for video relighting.

**Strengths:**

The paper is overall well written and the contributions are easy to understand. Video relighting is a challenging problem, where temporal flickering is a serious problem that is difficult to quantify and difficult to address. The proposed techniques are lightweight.

**Weaknesses:**

The contributions look like a concatenation of exceedingly simple techniques collected from different sources into a single framework for video relighting. Individually, each contribution is not particularly novel. Specifically,

1. The light stability score (section 3.1.1) is an ad-hoc procedure based on brightness thresholding, which leads to three time-series signals that assess the video's light fluctuation. Overall it looks very ad-hoc (not grounded in theoretical analysis) and overly simplistic.

2. To quantify detail preservation (section 3.1.2), Structural Similarity Index (SSIM) is simply adopted, a very well known image quality assessment metric. Equations (1) are (2) are standard SSIM. There is no novelty here.

3. HMA-LSF (section 3.2.2) is a simple combination of optimal flow and bilateral filter (a primitive edge-preserving filter that dates back to 1990s). The blending operation in equation (7) is simplistic and does not vary across the frame (i.e., the same $\alpha$ is used throughout).

4. LAB-DF (section 3.2.3) is a simple linear blending operation in equation (9).

**Questions:**

1. What are the key novelty for individual components, HMA-LSF and LAB-DF, above and beyond existing works, in flicker removal and texture restoration, respectively?

2. How sensitive is the HMA-LSF to the accuracy of the optical flow estimation? Bilateral filter is a combination of domain and rang filters. How are the two filters adjusted (via parameters $\sigma$'s in the Gaussian kernel) for optimal performance? Is parameter $\alpha$ in equation (7) automatically adjusted in a data-driven manner, or hand-tuned beforehand?

3. How is $\beta$ in equation (9) optimized?

---

> ### Author Response · Authors · 2025-11-23
> **Discussion with Reviewer frAN - Part 1**
>
> We sincerely thank the reviewers for their constructive comments and the time spent evaluating our paper. Please see our responses below.
>
> **W1: The light stability score (section 3.1.1) is an ad-hoc procedure based on brightness thresholding, which leads to three time-series signals that assess the video's light fluctuation. Overall it looks very ad-hoc (not grounded in theoretical analysis) and overly simplistic.**
>
> We respectfully disagree and argue that simplicity, when validated by strong empirical experiments, is a robust metric design. We address the concerns regarding theoretical grounding and validity below:
>
> While the mathematical formulation is straightforward, it is strictly **grounded in human perception**. As detailed in Appendix A.4, we conducted a blind human survey where our $S_{LS}$ achieved a perfect Spearman’s rank correlation 1 with human rankings of stability.
>
> To further address the reviewer's concern about the "ad-hoc" nature of our metric, we have newly evaluated our method using the **VBench benchmark**, a generic video generation benchmark. The VBench results for temporal flickering consistency show counter-intuitive results in which the unstable relit videos have even better temporal flickering consistency than the original videos. This makes our Light Stability Score more essential for the video relighting task.
>
> We also acknowledge the concern about the brightness threshold ($\tau$). However, our **sensitivity analysis** in Appendix A.3 demonstrates that the ranking of methods remains consistent across a wide range of thresholds. This confirms that the metric detects a fundamental instability in the video data and is not an artifact of parameter tuning.
>
> **W2: To quantify detail preservation (section 3.1.2), Structural Similarity Index (SSIM) is simply adopted, a very well known image quality assessment metric. Equations (1) are (2) are standard SSIM. There is no novelty here.**
>
> We wish to clarify that we do not claim the SSIM metric itself as a novel contribution. Rather, our contribution is the establishment of the **first principled evaluation paradigm tailored for video relighting**, a task that does not have a standardized assessment protocol. Previous works (e.g., Light-A-Video [1], TC-Light [2]) rely on generative metrics like Fréchet Inception Distance (FID) or CLIP scores. While useful for general image generation, these metrics fail to capture the detail degradation and light flickering problems. We deliberately chose SSIM because video relighting is a restoration and enhancement task, and SSIM does not need a GPU to compute.
>
> In addition, our newly added VBench evaluation results in Appendix B.5 that justify the necessity of our evaluation paradigm.
>
>
> **W3: HMA-LSF (section 3.2.2) is a simple combination of optimal flow and bilateral filter (a primitive edge-preserving filter that dates back to 1990s). The blending operation in equation (7) is simplistic and does not vary across the frame (i.e., the same
>  is used throughout).**
>
> The reviewer comments that our contributions are "simple combinations" of existing tools. We respectfully argue that simplicity in design, when grounded in a deep understanding of the problem, is a strength rather than a weakness. We moved away from the trend of "brute force" solutions that rely on increasingly large datasets or more sophisticated architectures. Instead, we investigated the root causes of failure in the existing video relighting model: light flickering and detail degradation, then designed our framework to mitigate the problems.
> Just as foundational mechanisms like the skip connections (summation of weights) or attention, our design is mathematically straight-forward yet very effective. While optical flow and bilateral filters are established tools, our novelty lies in how they are integrated to solve the specific artifacts of existing challenges. We are the first to successfully combine these traditional computer vision principles to resolve the light flickering and detail loss in recent SOTA models [1, 2].
>
>
> **W4: LAB-DF (section 3.2.3) is a simple linear blending operation in equation (9).**
>
> As with other foundational techniques, weighted sum operations are ubiquitous in modern work precisely due to their effectiveness. We argue that a straightforward solution that yields excellent empirical results is preferable to over-engineered alternatives. Complicating the merging process would increase computational requirements without providing significant performance gains.

---

> ### Author Response · Authors · 2025-11-23
> **Discussion with Reviewer frAN - Part 2**
>
> **Q1: What are the key novelty for individual components, HMA-LSF and LAB-DF, above and beyond existing works, in flicker removal and texture restoration, respectively?**
>
> This question is similar to weaknesses 3 and 4. In alignment with the **ICLR’s central belief on learning representations**, our work introduces specific technical novelties that **better represent and disentangle the components of video relighting**. For flicker removal, HMA-LSF improves temporal representations by effectively resolving the trade-off between stability and motion blur. For texture restoration, the LAB-DF module tackles the detail degradation characteristic of diffusion models by decoupling structural content from illumination. LAB-DF extracts illumination information and transfers it to the original high-fidelity input, ensuring robust structure preservation.
>
>
> **Q2: How sensitive is the HMA-LSF to the accuracy of the optical flow estimation? Bilateral filter is a combination of domain and rang filters. How are the two filters adjusted (via parameters 's in the Gaussian kernel) for optimal performance? Is parameter
>  in equation (7) automatically adjusted in a data-driven manner, or hand-tuned beforehand?**
>
> **We have newly added an ablation study (Appendix B.1)** to investigate the sensitivity of $\alpha$ in equation (7).  In short, as alpha increases, the light stability scores increase, but the motion trailing problem may occur, which is generally eliminated by LAB-DF later.
>
> In our implementation, the bilateral filter parameters are set to fixed values to provide a consistent baseline of edge-preserving smoothing. Its main task is to mitigate any possible remaining noise. Therefore, rather than adjusting the Gaussian kernels dynamically, we hand-tuned the parameters to restrict the filter's application to specific highlight regions ($\sigma_{r}$=140;$\sigma_{s}$=40).
>
> As described in (Line 312-314), our default $\alpha$ is adaptive (data-driven). The $\alpha$ value is dynamically adjusted based on the magnitude of the motion vectors calculated by the optical flow. This ensures that in regions of fast motion where optical flow is most prone to errors, the system automatically reduces $\alpha$ to rely more on the current frame, preventing motion trailing and minimizing the impact of flow inaccuracies. In addition, we offered both fixed and adaptive modes in the scripts for the user.
>
>
> **Q3: How is β in equation (9) optimized?**
>
> The β in equation 9 is not an optimized term, but rather a hyper-parameter that is meant to be tweaked by the user. A sharp drop in SSIM and light stability around 0.4 and 0.5, therefore, we adopted 0.3 in our comparative experiments. We had a section in **Appendix A.1** that shows a plot and discussion of the effect of beta. **We newly included demo videos in the anonymous link and a more detailed ablation study on the β in Appendix B.2 in the updated version.**
>
> **List of changes for the updated paper:**
> 1. Included ablation studies on alpha in Equation (7) and beta in Equation (9), as well as evaluations on videos with different motion speeds.
>
> 2. Evaluated the models using VBench and metrics from prior works.
>
> 3. Uploaded an extensive set of demo videos to the anonymous GitHub page for comparison and ablation studies.
>
> 4. Added a "Limitations and Future Work" section.
>
> 5. Refined figure captions for greater detail.
>
> **Highlights of our scope and contributions:**
>
> 1. We tackle the critical challenges of detail loss and flickering prevalent in SOTA approaches [1, 2].
>
> 2. Our method stands as the only current solution for high-resolution video relighting.
>
> 3. We introduce a new evaluation paradigm explicitly designed for this task, empirically demonstrating that it provides a more accurate and robust assessment of relighting quality.
>
> **References**:
>
> [1] Light-A-Video: Training-free Video Relighting via Progressive Light Fusion. (ICCV 2025)
>
> [2] TC-Light: Temporally Coherent Generative Rendering for Realistic World Transfer (NeurIPS 2025)

---

### Comment · Area_Chair_yjXp · 2025-11-26

Dear reviewers,

Please check the author's reply. Feel free to raise any questions or start a discussion, regardless of whether you will change the score.

Your AC.

---

### Meta-Review · Area_Chair_Gj9C · 2026-01-04

**Summary:**

The dominant concern driving the reject recommendations is contribution and evaluation validity: reviewers view the method/metric as a composition of simple/off-the-shelf components with limited novelty (frAN, afb4), and more critically argue the evaluation paradigm is incomplete or potentially biased toward temporal consistency and similarity to the input, without directly measuring relighting quality (realism / physical plausibility/strength of illumination change) (afb4/C9X2/AhoR).

**Reviewer Concerns:**

## Addressed

**Missing evidence (C9X2, afb4, AhoR):**
Reviewers requested stronger evidence to support the paper’s claims. The authors added multiple demo videos and additional qualitative comparisons (including cases with larger camera motion). This partially addresses the concern by improving empirical support and interpretability.

**Missing existing metrics / fair comparison (C9X2, afb4):**
Reviewers asked for evaluations using established metrics to enable fairer comparisons. The authors added results with VBench and other commonly used metrics (e.g., FID/CLIP-related scores) in the appendix.

**Robustness to motion & optical-flow sensitivity (AhoR, C9X2, frAN, afb4):**
Authors added motion-speed studies and ablations on key parameters (e.g., $\alpha$ in Eq.7) and described an adaptive $\alpha$ scheme based on flow magnitude; also included robustness analysis.

**Hyperparameter sensitivity / tuning questions (frAN, C9X2, afb4):**
Authors added ablations for $\alpha$ and $\beta$, clarified $\beta$ as a user-tunable hyperparameter with empirical choice, and stated hyperparameters are fixed globally (not per-scene tuned).

**Runtime comparisons (C9X2, afb4):**
Authors added runtime comparisons in Appendix B.4.



## Unsolved

**Evaluation paradigm does not measure “relighting quality” (afb4; partially echoed by others).**
Despite adding VBench/prior metrics and human study, Reviewer afb4 maintains that the proposed paradigm (SSIM + Light Stability Score) fundamentally emphasizes temporal consistency and similarity to input, and could reward minimal relighting rather than high-quality illumination change. The authors argue that (i) artifact suppression is a prerequisite and (ii) aesthetics/physics metrics are open problems and are covered via human study + VBench. However, afb4 does not accept this framing, and this remains the key unresolved issue.

**Novelty (frAN, partially afb4).**
Reviewers argued that the method largely combines well-known components (e.g., optical flow + bilateral filtering, linear LAB fusion, threshold-based stability scoring) and that the novelty of each individual component is limited. The rebuttal frames simplicity as a strength and argues the integration is effective for high-resolution relighting. Still, the concern remains that the pipeline does not introduce sufficiently new technical insight beyond expected effects from standard operations. Moreover, the proposed metric lacks novelty.

**Scope limitation: fine-grained, physically grounded relighting control (AhoR, C9X2, afb4).**
Authors clearly position the work as text-driven generative relighting rather than physics-based relighting, but questions remain whether the results constitute “relighting” vs “stylization,” and whether hard shadows/highlights are meaningfully controllable/evaluable. The authors acknowledge these limitations; they are not fully resolved.

**Unclarity:**
The authors improved captions/figures, added a Limitations section, and clarified optical-flow choices/implementation. However, the paper still has notable readability obstacles in places (e.g., the pipeline illustration in Figure 3 remains difficult to parse), which may hinder reproducibility and understanding.

**Reviewer Scores:**

Reviewer frAN (2 $\rightarrow$ 2):
Authors added parameter sensitivity analysis and clarified adaptivity/hand-tuning; improved justification for SLS via human correlation and threshold sensitivity. However, frAN’s core critique is low novelty (“concatenation of simple techniques”) and “ad-hoc metric,” which is only partially mitigated.

Reviewer AhoR (6$\rightarrow$ 6):
Main concerns (motion robustness; light direction control evidence) were addressed via motion-speed ablation + demos + clarification of coarse direction control.

Reviewer C9X2 (2$\rightarrow$ 2):
Although many concrete requests were addressed: VBench + prior metrics, runtime comparison, optical-flow details/ablation, more demos, and limitations, there are still several concerns were remain. I particular, the method figure is difficult to understand, and the application scope is limited due to the method’s inability to properly respect physical constraints.

Reviewer afb4 (2$\rightarrow$ 2):
Reviewer explicitly states after rebuttal that their primary concern remains fundamentally unaddressed (evaluation paradigm lacks relighting-quality assessment; potentially biased).

---

### Decision · Program_Chairs · 2026-01-26

Reject